# PURE: Prompt Evolution with Graph ODE for Out-of-distribution Fluid Dynamics Modeling

**Hao Wu**[1,3], **Changhu Wang**[2], **Fan Xu**[1], **Jinbao Xue**[3],
**Chong Chen**[4], **Xian-Sheng Hua**[4], **Xiao Luo**[2,*]

[1]University of Science and Technology of China, [2]University of California, Los Angeles,
[3]Tencent , [4]Terminus Group

wuhao2022@mail.ustc.edu.cn, wangch156@g.ucla.edu, markxu@mail.ustc.edu.cn
markxu@mail.ustc.edu.cn,jinbaoxue@tencent.com, chenchong.cz@gmail.com,
huaxiansheng@gmail.com, xiaoluo@cs.ucla.edu

## Abstract

This work studies the problem of out-of-distribution fluid dynamics modeling. Previous works usually design effective neural operators to learn from mesh-based data structures. However, in real-world applications, they would suffer from distribution shifts from the variance of system parameters and temporal evolution of the dynamical system. In this paper, we propose a novel approach named Prompt Evolution with Graph ODE (PURE) for out-of-distribution fluid dynamics modeling. The core of our PURE is to learn time-evolving prompts using a graph ODE to adapt spatio-temporal forecasting models to different scenarios. In particular, our PURE first learns from historical observations and system parameters in the frequency domain to explore multi-view context information, which could effectively initialize prompt embeddings. More importantly, we incorporate the interpolation of observation sequences into a graph ODE, which can capture the temporal evolution of prompt embeddings for model adaptation. These time-evolving prompt embeddings are then incorporated into basic forecasting models to overcome temporal distribution shifts. We also minimize the mutual information between prompt embeddings and observation embeddings to enhance the robustness of our model to different distributions. Extensive experiments on various benchmark datasets validate the superiority of the proposed PURE in comparison to various baselines. Our codes are available at https://github.com/easylearningscores/PURE_main.

## 1 Introduction

Fluid dynamics [44, 89] is a critical area in the field of mechanics and computational fluid dynamics has emerged as a powerful tool to understanding fluid flow [32, 58, 67, 45]. Recently, various machine learning approaches have been widely adopted to solve the problem in a data-driven manner [59, 46, 66, 65, 18, 7, 81], which can achieve high efficiency in comparison to previous traditional numerical solvers. Moreover, they enjoy strong applicability when the underlying rules are not explicit, such as real-world weather forecasting [4] and disease transmission [68].

In literature, existing data-driven fluid dynamics modeling approaches can be roughly divided into grid-based approaches [12, 17] and geometry-based approaches [59, 66, 65, 20]. Grid-based approaches construct regular meshes and then utilize neural operators to explore spatio-temporal relationships. In contrast, geometry-based approaches focus on irregular point clouds and then utilize graph neural networks (GNNs) [30, 70] to learn from the interaction between mesh points.

---

[*]Corresponding author.

38th Conference on Neural Information Processing Systems (NeurIPS 2024).

Despite their great success, existing approaches [88, 25] generally assume that training and test data share the same data distribution [59, 66, 65, 47], which could be not the case in real-world applications. In particular, there are two typical types of distribution shifts in dynamical systems, i.e., parameter-based shifts, and temporal distribution shifts. Firstly, different dynamical systems could involve different parameters in underlying rules, such as coefficients in PDEs and pressures in fluid systems [3, 65]. Secondly, during long-term auto-regressive forecasting, the input data distribution could vary hugely during the temporal evolution [91]. As in previous works [22, 33], machine learning approaches usually suffer from huge performance degradation when it comes to distribution shifts. Therefore, in this paper, we focus on the problem of out-of-distribution fluid dynamics modeling to enhance the performance under potential distribution shifts.

In this paper, we propose a new approach named $\underline{P}$rompt Evol$\underline{u}$tion with G$\underline{r}$aph OD$\underline{E}$ (PURE) for out-of-distribution fluid dynamics modeling. The high-level idea of our proposed PURE is to adapt well-trained forecasting approaches to different out-of-distribution scenarios by learning time-evolving prompts [51, 84, 9]. To begin, we extract multi-view context signals from both historical observations and system parameters in the frequency domain using the attention mechanism, which can effectively initialize prompt embeddings under parameter-based shifts. More importantly, to capture temporal distribution shifts, we combine the interpolation of observation sequences into a graph ODE framework, which can utilize the interaction between prompt embeddings and observation embeddings for high-quality time-evolving prompt embeddings. Then, we concatenate our prompt embeddings and observation embeddings for model adaptation and enhance the robustness of our PURE to distribution variance by minimizing their mutual information using adversarial learning. Extensive experiments on a range of fluid dynamics datasets validate the superiority of the proposed PURE in comparison to various state-of-the-art approaches.

In summary, the contribution of our paper can be summarized as follows: (1) *Problem Connection.* We are the *first* to connect prompt learning with dynamical system modeling to solve the issue of out-of-distribution shifts. (2) *Novel Methodology.* Our PURE first learns from historical observations and system parameters to initialize prompt embeddings and then adopts a graph ODE with the interpolation of observation sequences to capture their continuous evolution for model adaptation under out-of-distribution shifts. (3) *Superior Performance.* Comprehensive experiments validate the effectiveness of our PURE in different challenging settings.

## 2  Problem Setup

Given a fluid dynamical system, we have $N$ sensors within the domain $\Omega$, with their locations denoted as $\boldsymbol{x}_1, \cdots, \boldsymbol{x}_N$, where $\boldsymbol{x}_i \in \mathbb{R}^{d_l}$. The observations at time step $t$ are represented as $\boldsymbol{s}_1^t, \cdots, \boldsymbol{s}_N^t$, where $\boldsymbol{s}_i^t \in \mathbb{R}^{d_o}$ and $d_o$ indicates the number of observation channels. Dynamical systems are governed by underlying system rules, such as PDEs with coefficient $\boldsymbol{\xi}$. Variations in system parameters may lead to different environments, potentially resulting in distribution shifts [54, 80, 6, 27]. In our study, we are provided with historical observation sequences $\{\boldsymbol{s}_i^{1:T_0}\}_{i=1}^N$ and physical parameters $\boldsymbol{\xi}$ (e.g., coefficients in the PDEs). Our goal is to predict the future observations of each sensor $\boldsymbol{s}_i^{T_0+1:T_0+T}$. In dynamical systems, the out-of-distribution problem examines model performance when predicting under unseen parameter distributions or environments. Let $\boldsymbol{u}^t = [\boldsymbol{s}_1^t, \cdots, \boldsymbol{s}_N^t]$, these systems evolve according to $\frac{d\boldsymbol{u}}{dt} = F(\boldsymbol{u}, \boldsymbol{\xi})$, where $\boldsymbol{u}$ represents the observations and $\boldsymbol{\xi}$ denotes the system parameters. When $\boldsymbol{\xi} \sim P(\boldsymbol{\xi})$, the state trajectory $\boldsymbol{u}^{1:T_0}$ follows the distribution $P(\boldsymbol{u}^{1:T_0}|\boldsymbol{\xi})$. Assume we learn a learned mapping function $f$ from $\boldsymbol{u}^{1:T_0}$ to $\boldsymbol{u}^{T_0+1:T_0+T}$, i.e., $\boldsymbol{u}^{T_0+1:T_0+T} = f(\boldsymbol{u}^{1:T_0})$ and there could be different distributions across training and test datasets, i.e., $P_{\text{train}}(\boldsymbol{\xi}) \neq P_{\text{test}}(\boldsymbol{\xi})$, which results in $P_{\text{train}}(\boldsymbol{u}^{1:T_0}) \neq P_{\text{test}}(\boldsymbol{u}^{1:T_0})$. Moreover, when conducting rollout prediction, we are required to feed the output back to the model, i.e., $\boldsymbol{u}^{T_{start}:T_{start}+T-1} = f(\boldsymbol{u}^{T_{start}-T_0:T_{start}-1})$, with $P(\boldsymbol{u}^{1:T_0}|\boldsymbol{\xi}) \neq P(\boldsymbol{u}^{T_{start}-T_0:T_{start}-1}|\boldsymbol{\xi}, T_{start})$.

## 3  The Proposed PURE

### 3.1  Motivation and Framework Overview

This paper addresses the challenge of out-of-distribution fluid system modeling, which is complicated by parameter-based and temporal distribution shifts. Specifically, our function $f(\cdot)$ can suffer from

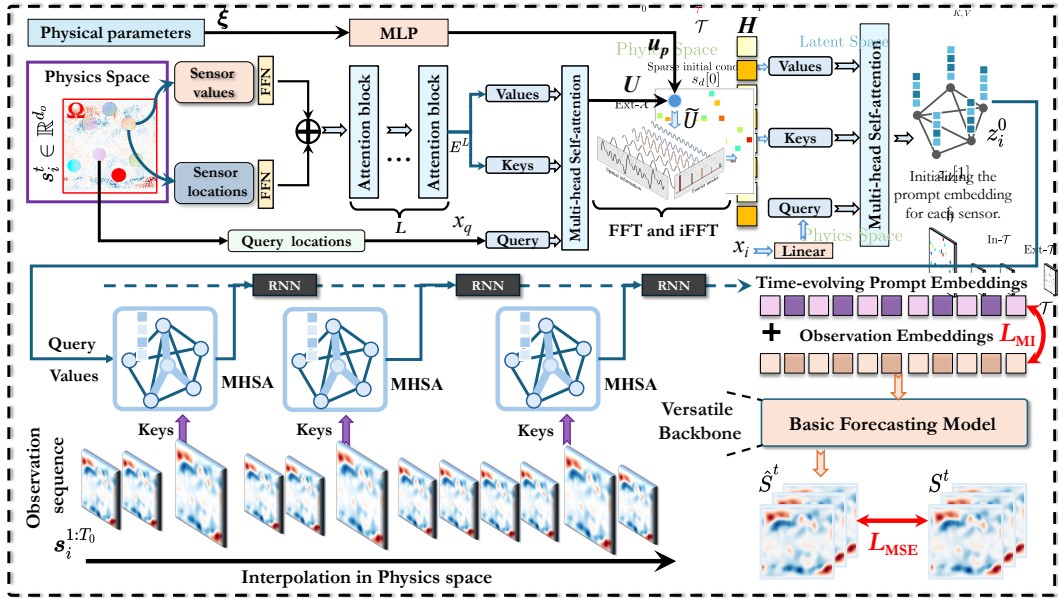

Figure 1: Overview of the PURE framework.

a serious distribution shift result from different $\boldsymbol{\xi}$ and $T_{start}$, i.e., $P(\boldsymbol{u}_{input}|\boldsymbol{\xi}, T_{start})$. To reduce the impact of distribution shift, we aim to learn invariant observation embeddings $\boldsymbol{\mu}^t$ to different environments, i.e., $\boldsymbol{\xi}$ and $T_{start}$ for better generalization and utilize prompt embeddings $\boldsymbol{z}^t$ to indicate the current environment for final prediction. In formulation, we have:

$$\boldsymbol{z}^t \perp \boldsymbol{\mu}^t, u_{output} = \phi([\boldsymbol{\mu}^t, \boldsymbol{z}^t]). \tag{1}$$

The first term ensures the invariance of observation embeddings by decoupling observation embeddings and prompt embeddings. The second term aims to combine both two embeddings to generate the future predictions. Therefore, we propose a novel approach named PURE as:

$$\boldsymbol{\mu}^t = \text{BasicModel}(\boldsymbol{u}_{input}), \quad \boldsymbol{z}^0 = \text{ContextMining}(\boldsymbol{u}_{input}), \quad \boldsymbol{z}^t = \text{GraphODE}(\boldsymbol{z}^0, t). \tag{2}$$

where a basic model is adopted to generate observation, and we adopt context mining and graph ODE to learn time-varying prompt embeddings. Given a basic forecasting model (Eqn. 2), our PURE contains three key modules: (1) *Multi-view Context Exploration*, which explores spatio-temporal data using both the attention mechanism and the frequency domain to initialize prompt embeddings (Eqn. 2). (2) *Time-evolving Prompt Learning*, which incorporates the interpolation of observation sequences into a graph ODE to learn the evolution of prompt embeddings (Eqn. 2). (3) *Model Adaptation with Prompt Embeddings*, which leverages the time-evolving prompts to mitigate the temporal distribution shifts in fluid dynamics models (Eqn. 1). More details are in Figure 1.

## 3.2 Multi-view Context Exploration from Spatio-temporal Data

The main idea of our PURE is to utilize prompt learning to solve the issue of out-of-distribution shifts [54, 80, 6, 27] in fluid dynamical systems. Prompt learning [51, 84, 9] is an effective manner to adapt language models to various downstream tasks. In our scenarios, we aim to learn from both historical spatio-temporal information and system parameters to initialize our prompt embeddings, which can effectively solve the parameter-based distribution shifts. Here, we first follow the attention mechanism [69, 86, 10, 50] to reconstruct the field value and then adopt the Fourier neural operator [45] to integrate multi-view context information.

In particular, given each location, we map each location $\boldsymbol{x}_i$ and each initial observation $\boldsymbol{s}_i^0$ into a position embedding $\boldsymbol{p}_i$ and an observation embedding $\boldsymbol{q}_i$ using two feed-forwarding networks (FFNs) $\phi^{PE}(\cdot)$ and $\phi^{OE}(\cdot)$, and then aggregate $\boldsymbol{p}_i$ and $\boldsymbol{q}_i$ using the Hadamard product followed by stacking $L$ self-attention blocks for representation learning. In formulation,

$$\boldsymbol{e}_i = \boldsymbol{p}_i \odot \boldsymbol{q}_i, \boldsymbol{E}^{l+1} = \phi^{SA,(l)}(\boldsymbol{E}^l), \tag{3}$$

where $\odot$ denotes the Hadamard product, $\boldsymbol{E}^0$ is constructed by stacking $\{\boldsymbol{e}_i\}_{i=1}^N$, and $\phi^{SA,(l)}$ is the self-attention block at the layer $l$. Afterward, we adopt the attention mechanism [69] to retrieve the representations for each query position $\boldsymbol{x}_q$ as:

$$\boldsymbol{u}^q = \mathrm{softmax}\left(\frac{[\boldsymbol{W}^Q\phi^{PE}(\boldsymbol{x}^q)]^T \cdot [\boldsymbol{W}^K\boldsymbol{E}^L]}{\sqrt{d}}\right) \cdot \boldsymbol{W}^V\boldsymbol{E}^L, \tag{4}$$

where $\boldsymbol{W}^*$ is a learnable weight matrix for feature transformation and $d$ is the hidden dimension. By retrieving the representations at each regular grid, we generate the 3D representation tensor $\boldsymbol{U}$. Each tensor would be concatenated with the parameter embedding $\boldsymbol{u}^p = \phi^{PA}(\boldsymbol{\xi})$ for multi-view information integration, which results in the final tensor $\tilde{\boldsymbol{U}}$. Then, we utilize the frequency domain for representation enhancement to generate a prompt tensor $\boldsymbol{H}$. Here, we first transfer the tensor into the frequency domain using a Fast Fourier Transformer (FFT) operator [45] and then adopt an FFN for feature transformation. Lastly, an inverse Fast Fourier Transformer (iFFT) operator is adopted to convert the features back to the spatial domain. Formally,

$$\boldsymbol{H} = \mathrm{iFFT}(\mathrm{FFN}(\mathrm{FFT}(\tilde{\boldsymbol{U}}))), \tag{5}$$

where $\mathrm{FFT}(\cdot)$ and $\mathrm{iFFT}(\cdot)$ denote the FFT and iFFT operators, respectively. Since the input of spatio-temporal models would be irregular, we flatten the tensor $\boldsymbol{H}$, and retrieve prompts for each sensor from the prompt tensor using:

$$\boldsymbol{z}_i^0 = \mathrm{softmax}\left(\frac{[\boldsymbol{W}^{Q'}\phi^{PE}(\boldsymbol{x}_i)]^T \cdot [\boldsymbol{W}^{K'}flatten(\boldsymbol{H})]}{\sqrt{d}}\right) \cdot \boldsymbol{W}^{V'}flatten(\boldsymbol{H}), \tag{6}$$

where $flatten(\cdot)$ is a flattening operator to transform 3D tensors to 2D matrices. Through the frame reconstruction, we can extract important spatio-temporal signals from the frequency domain, which is effective in initializing the prompt embedding for each sensor.

## 3.3 Time-evolving Prompt Learning with Graph ODE

To capture temporal distribution shifts within one system, static prompt embeddings [51] from context exploration are far from satisfactory. Our solution is to obtain continuous time-evolving prompts at any timestamp. To achieve this, we view the output of Eqn. 6 as the initial prompt embeddings and then incorporate the attention mechanism into a continuous graph ODE, which combines the interpolations of observations with the graph structure to learn the evolution of prompt embeddings.

In particular, given the initial prompt embeddings, we introduce two functions $\psi_a(\cdot)$ and $\psi_r(\cdot)$ for relation mining and feature aggregation. $\psi_r(\cdot)$ calculates the interaction between the centroid node and each of its neighboring nodes and $\psi_a(\cdot)$ aggregates all the neighborhood interactions to determine the evolution. Therefore, a graph ODE can be formulated by the following formulation:

$$\frac{d\boldsymbol{z}_i^t}{dt} = \psi_a\left(\sum_{j\in\mathcal{S}^t(i)} \psi_r([\boldsymbol{z}_i^t, \boldsymbol{z}_j^t])\right), \tag{7}$$

where $\mathcal{S}^t(i)$ collects the sensors from the neighbours of $i$ at timestamp $t$. However, Eqn. 7 neglects observations themselves during evolution, which are directly related to temporal distribution shifts in dynamical systems. Thus, it could generate suboptimal prompt embeddings. To tackle the issue, we conduct the interpolations of observation sequence $\boldsymbol{s}_i^{1:T_0}$, which results in $\boldsymbol{s}_i^t$ at any timestamp. Then, we incorporate them into our graph ODE using the attention mechanism by rewriting Eqn. 7 into:

$$\frac{d\boldsymbol{z}_i^t}{dt} = \psi_a\left(\sum_{j\in\mathcal{S}^t(i)} \mathrm{softmax}\left(\frac{[\tilde{\boldsymbol{W}}^Q\boldsymbol{z}_i^t]^T \cdot [\tilde{\boldsymbol{W}}^K\boldsymbol{s}_j^t]}{\sqrt{d}}\right) \cdot \psi_r([\boldsymbol{z}_i^t, \boldsymbol{z}_j^t])\right). \tag{8}$$

where $\tilde{\boldsymbol{W}}^Q$ and $\tilde{\boldsymbol{W}}^K$ are two matrices to generate the query and key, respectively. Here, we utilize the prompt embeddings and interpolated observations to serve as the query and the key. In this way, we effectively model their interaction to adjust the derivative in the graph ODE, which can help generate proper prompt embeddings for our model adaptation under temporal distribution shifts.

### 3.4 Model Adaptation with Prompt Embeddings

Finally, we incorporate our prompt embedding into our basic spatio-temporal forecasting model, and then introduce the optimization objective for the end-to-end training.

**Basic Forecasting Model.** Our time-evolving prompt embeddings can be easily incorporated into any spatio-temporal forecasting model. To make the best of our efficacy, we utilize a simple yet powerful basic model as our default model and also explore the performance of our PURE on more existing forecasting models. The input of our model is the observations of sensors from between the interval $[1, T_0]$ and output the predictions in $[T_0 + 1, T_0 + T]$, i.e., $\boldsymbol{S}^{1:T_0} \rightarrow \boldsymbol{S}^{T_0, T_0+T}$ where $\boldsymbol{S}^*$ is stacked by $\boldsymbol{s}_i^*$. In particular, our basic model first generates the embeddings of different observations, and then reconstructs the irregular observations into frames on grids using the reconstruction modules in Sec. 3.2. More importantly, we introduce two parallel modules, i.e., a Fourier neural operator [45] and a ViT-based convolution [19] and extract complementary feature maps [81], which would be fused to generate the predicted frames in the future. More details of our basis forecasting model can be found in Appendix B. Additionally, we also use other basic models.

**Adaptation and Optimization.** Note that we generate observation embeddings in our basic module, i.e., $\boldsymbol{\mu}_i^t = \phi^{enc}(\boldsymbol{s}_i^t)$. To adapt our model under distribution shifts, we concatenate the observation embeddings and prompt embeddings into updated embeddings $\tilde{\boldsymbol{\mu}}_i^t$ as follows:

$$\tilde{\boldsymbol{\mu}}_i^t = [\boldsymbol{\mu}_i^t, \boldsymbol{z}_i^t], \tag{9}$$

which will be fed into the subsequent modules in the basic model. To optimize the whole framework, we first minimize the mean squared error (MSE) between the predicted observation and the ground truth as follows:

$$\mathcal{L}_{MSE} = \sum_{t=T_0+1}^{T_0+T} ||\hat{\boldsymbol{S}}^t - \boldsymbol{S}^t||, \tag{10}$$

where $\hat{\boldsymbol{S}}^t$ denotes our predicted observation for every node and $\boldsymbol{S}^t$ denotes the ground truth observations. Moreover, to enhance the invariance of our model to different scenarios, we turn to invariant learning [72, 42, 77] to decouple various prompt embeddings and observation embeddings, which promotes the observation embeddings to be less sensible to different distributions. To achieve this, we minimize the mutual information between observation embeddings and observation embeddings, i.e., $I(\boldsymbol{\mu}_i^t; \boldsymbol{z}_i^t)$. In our work, we adopt a Jensen-Shannon mutual information estimator [40, 53] $T_\gamma(\cdot, \cdot)$ where $\gamma$ denotes the parameters to estimate their mutual information. Then, we collect all the corresponding pairs of $(\boldsymbol{\mu}_i^t, \boldsymbol{z}_i^t)$ using $\mathcal{P}$ and all the possible pairs of $(\boldsymbol{\mu}_i^t, \boldsymbol{z}_j^t)$ using $\mathcal{N}$. The adversarial learning objective can be written as:

$$\mathcal{L}_{MI} = max_{\gamma'}\{\frac{1}{|\mathcal{P}|}\sum_{(\boldsymbol{\mu}_i^t, \boldsymbol{z}_i^t) \in \mathcal{P}} sp(-T_{\gamma'}(\boldsymbol{\mu}_i^t, \boldsymbol{z}_i^t)) + \frac{1}{|\mathcal{N}||\mathcal{P}|}\sum_{(\boldsymbol{\mu}_i^t, \boldsymbol{z}_j^t) \notin \mathcal{P}} -sp(-T_{\gamma'}(\boldsymbol{\mu}_i^t, \boldsymbol{z}_j^t))\}, \tag{11}$$

in which $sp(\boldsymbol{x}) = \log(1 + e^{\boldsymbol{x}})$ represents the softplus function. In summary, the overall objective can be written as:

$$\mathcal{L} = \mathcal{L}_{MSE} + \lambda \mathcal{L}_{MI}, \tag{12}$$

where $\lambda$ is a coefficient to balance two loss objectives. The algorithm is summarized in Appendix D.

### 3.5 Theoretical Analysis

In this part, we provide a theoretical analysis to demonstrate how PURE works. Our focus is primarily on theoretically showing the necessity of incorporating the observations themselves during evolution. For simplicity of analysis, we assume that Eqn. 7 can be rewritten as:

$$\frac{d\boldsymbol{z}_i^t}{dt} = \frac{1}{\#(\mathcal{S}^t(i))}\sum_{j \in \mathcal{S}^t(i)}(M_1\boldsymbol{z}_i^t + M_2\boldsymbol{z}_j^t) = M_1\boldsymbol{z}_i^t + \frac{1}{\#(\mathcal{S}^t(i))}\sum_{j \in \mathcal{S}^t(i)}M_2\boldsymbol{z}_j^t, \tag{13}$$

where $\#(\cdot)$ caluclates the size of the set. For the sake of simplicity in the proof, we assume that $\boldsymbol{z}_i^t$ is one-dimensional and consider only the ODE above for $i$ (not the entire system of ODEs). Then, Eqn. 13 can be rewritten as:

$$\frac{dz_i^t}{dt} = \frac{1}{\#(\mathcal{S}^t(i))}\sum_{j \in \mathcal{S}^t(i)}(M_1z_i^t + M_2z_j^t) = M_1z_i^t + b(t), \tag{14}$$

where $b(t)$ is a function. To characterize temporal distribution shifts, we assume that a portion of the corresponding true $z_j^t$ has a constant shift. With the potential environmental change, Eqn. 13 can be rewritten as:

$$\frac{dz_i^t}{dt} = \frac{1}{\#(\mathcal{S}^t(i))} \sum_{j \in \mathcal{S}^t(i)} (M_1 z_i^t + M_2 z_j^t) = M_1 z_i^t + b'(t), \tag{15}$$

where $|b(t) - b'(t)| \geq c_0$, suggesting the constant shift $c_0$.

**Theorem 3.1.** *Given the following ODEs in $\mathbb{R}$,*

$$\begin{aligned}
\dot{x} &= M_1 x + b(t), & x(0) &= x_0, \\
\dot{y} &= M_1 y + b'(t), & y(0) &= x_0,
\end{aligned} \tag{16}$$

*where $|b(t) - b'(t)| \geq c_0$, there exists a positive constant $c_1$ such that*

$$|x(t) - y(t)| \geq c_1(e^{M_1 t} - 1), \text{ for all } t > 0. \tag{17}$$

The proof of Theorem 3.1 can be found in Appendix A. Theorem 3.1 suggests that even with the simplest one-dimensional linear ODE, significant differences in the solutions will arise if temporal distribution shifts are neglected. Next, we will focus on how Eqn. 8 addresses this issue. In this case, we assume that

$$\psi_r([z_i^t, z_j^t]) = M_1 z_i^t + M_2 z_j^t. \tag{18}$$

Then, Eqn. 7 can be written as:

$$\frac{dz_i^t}{dt} = \psi_\alpha\left(\sum_{j \in \mathcal{S}^t(i)} (M_1 z_i^t + M_2 z_j^t)\right) = \psi_\alpha\left(M_1 z_i^t + \frac{1}{\#(\mathcal{S}^t(i))} \sum_{j \in \mathcal{S}^t(i)} M_2 z_j^t\right) = \psi_\alpha\left(M_1 z_i^t + b(t)\right), \tag{19}$$

where

$$b(t) = \frac{1}{\#(\mathcal{S}^t(i))} \sum_{j \in \mathcal{S}^t(i)} M_2 z_j^t. \tag{20}$$

Similarly, Eqn. 8 can be written as:

$$\frac{dz_i^t}{dt} = \psi_\alpha\left(M_1 z_i^t + b'(t)\right), \tag{21}$$

where

$$b'(t) = \sum_{j \in \mathcal{S}^t(i)} \text{softmax}\left(\frac{[\tilde{\boldsymbol{W}}^Q \boldsymbol{z}_i^t]^T \cdot [\tilde{\boldsymbol{W}}^K \boldsymbol{s}_j^t]}{\sqrt{d}}\right) \cdot M_2 z_j^t. \tag{22}$$

For simplicity of notation, we omit the superscript $i$. Write $F(z, t) = \psi_\alpha\left(M_1 z + b(t)\right)$ and $G(z, t) = \psi_\alpha\left(M_1 z^t + b'(t)\right)$. Then, we have the following theorem with the proof in Appendix A.

**Theorem 3.2.** *Assume that the attention mechanism satisfies that $|b'(t) - b(t)| \leq \epsilon$, for all $t > 0$, and the function $\phi_\alpha$ is L-Lipschitz. Given the following ODEs in $\mathbb{R}$,*

$$\begin{aligned}
\dot{x} &= \psi_\alpha(M_1 x + b(t)) = F(x, t), & x(0) &= x_0, \\
\dot{y} &= \psi_\alpha(M_1 y + b'(t)) = G(y, t), & y(0) &= x_0,
\end{aligned} \tag{23}$$

*there exists two constants $c_2$ and $c_3$ such that*

$$|x(t) - y(t)| \leq \epsilon c_2(e^{c_3 t} - 1), \text{ for all } t > 0. \tag{24}$$

Thoerem 3.2 shows that as long as the attention mechanism is sufficiently good, we can approximate the true ODE with arbitrary precision using Eqn. 8, even in the presence of environmental change.

Table 1: We compare our study's performance with 10 baselines. We magnify the MSE of 3D-Reaction-Diffusion by 100 times. **Green** **Yellow** Red mean best, second, worst MSE.

| MODEL | BENCHMARKS | | | | | | | | | |
| --- | --- | --- | --- | --- | --- | --- | --- | --- | --- | --- |
| | PROMETHEUS | | NAVIER–STOKES | | SPHERICAL-SWE | | 3D REACTION–DIFF | | ERA5 | |
| | w/o OOD | w/ OOD | w/o OOD | w/ OOD | w/o OOD | w/ OOD | w/o OOD | w/ OOD | w/o OOD | w/ OOD |
| U-NET [64] | 0.0931 | **0.1067** | 0.1982 | 0.2243 | 0.0083 | 0.0087 | 0.0148 | 0.0183 | 0.0843 | 0.0932 |
| RESNET [21] | 0.0674 | 0.0696 | 0.1823 | 0.2301 | 0.0081 | 0.0192 | 0.0151 | 0.0186 | 0.0921 | 0.0977 |
| VIT [10] | 0.0632 | 0.0691 | **0.2342** | **0.2621** | 0.0065 | 0.0072 | 0.0157 | 0.0192 | 0.0762 | 0.0786 |
| SWINT [49] | 0.0652 | 0.0729 | 0.2248 | 0.2554 | 0.0062 | 0.0068 | 0.0155 | 0.0190 | 0.0782 | 0.0832 |
| FNO [45] | **0.0447** | **0.0506** | 0.1556 | 0.1712 | 0.0038 | 0.0045 | 0.0132 | 0.0179 | 0.7233 | **0.9821** |
| UNO [1] | 0.0532 | 0.0643 | 0.1764 | 0.1984 | 0.0034 | 0.0041 | **0.0121** | 0.0164 | 0.6652 | 0.7621 |
| CNO [63] | 0.0542 | 0.0655 | 0.1473 | 0.1522 | 0.0037 | 0.0038 | 0.0145 | 0.0182 | 0.5243 | 0.7821 |
| NMO [82] | 0.0397 | 0.0483 | **0.1021** | **0.1032** | **0.0026** | 0.0031 | 0.0129 | **0.0168** | **0.0432** | **0.0563** |
| CGODE [26] | 0.0761 | 0.0843 | 0.2035 | 0.2243 | **0.0873** | **0.0987** | **0.8371** | **0.9261** | **0.8721** | 0.9872 |
| DGODE [80] | **0.0344** | **0.0359** | **0.0805** | 0.0925 | **0.0022** | **0.0028** | 0.0122 | 0.0156 | 0.0543 | 0.0635 |
| OURS + PURE | **0.0323** | **0.0328** | **0.0752** | **0.0763** | **0.0022** | **0.0024** | **0.0119** | **0.0127** | **0.0398** | **0.0401** |
| PROMOTION | 6.10% | 8.63% | 6.58% | 26.07% | 0.00% | 16.67% | 1.65% | 22.56% | 7.87% | 28.77% |

Table 2: This table shows the performance of the PURE framework across different benchmarks.

| MODEL | BENCHMARKS | | | | | | | | | |
| --- | --- | --- | --- | --- | --- | --- | --- | --- | --- | --- |
| | PROMETHEUS | | NAVIER–STOKES | | SPHERICAL-SWE | | 3D REACTION–DIFF | | ERA5 | |
| | ORI | +PURE | ORI | +PURE | ORI | +PURE | ORI | +PURE | ORI | +PURE |
| RESNET [21] | 0.0674 | 0.0542 | 0.1823 | 0.1492 | 0.0081 | 0.0067 | 0.0151 | 0.0141 | 0.0921 | 0.0896 |
| NMO [10] | 0.0397 | 0.0281 | 0.1021 | 0.0876 | 0.0026 | 0.0012 | 0.0129 | 0.0123 | 0.0432 | 0.0389 |
| DGODE [49] | 0.0344 | 0.0201 | 0.0805 | 0.0792 | 0.0022 | 0.0020 | 0.0122 | 0.0110 | 0.0543 | 0.0462 |

## 4 Experiment

### 4.1 Experimental Settings

**Benchmarks.** We study Benchmarks from three domains, as shown in Table 5. ▷ **Computational Fluid Dynamics.** We use Prometheus [80] and follow the original setup for environment segmentation. ▷ **Real-world Data.** We employ the ERA5 [23], using different combinations of variables as the environment. In detail, we use ERA5 data with variables such as surface pressure (Sp), sea surface temperature (SST), sea surface height (SSH), and two-meter temperature (T2m) to predict temperature. ▷ **Partial Differential Equations.** The 2D Navier-Stokes equations [45] describe fluid motion, with the primary variable being the viscosity coefficient $\nu$, which quantifies internal friction in the fluid, simulating vorticity values under ten different viscosity coefficients. The spherical shallow water equations [14] simulate large-scale atmospheric and oceanic fluid motion on Earth's surface, also with viscosity coefficient $\nu$ as the main variable, involving tangential vorticity ($w$) and fluid thickness ($h$) on a spherical surface. The 3D reaction-diffusion equations describe the diffusion and reaction of chemicals in space [62], with the primary variable being the diffusion coefficient $D$, representing the rate of chemical diffusion in space, including $u, v$ velocity components. More details see Appendix E.

**Baselines.** We select representative models from three domains as baselines. ▷ **Visual Backbone Networks.** We include ResNet [21], U-Net [64], Vision Transformer(ViT) [10], and Swin Transformer(SWINT) [49]. ▷ **Neural Operator Architectures.** We cover FNO [45], UNO [1], CNO [63], and NMO [82]. ▷ **Graph-ODE Architectures.** We feature CG-ODE [26], and DGODE [80].

**Tasks.** We evaluate model performance for various prediction tasks through the following scenarios and use MSE as metrics. The specific tasks are as follows:

▷ **Generalization Experiments:** • **Out-of-Distribution Generalization:** We train the model In-Domain environment and test it in Adaptation environment to verify its generalization ability. • **Spatial Generalization & Temporal Generalization:** In the Prometheus, we train the model at 75% sparsity and test it at $s \in \{5\%, 25\%, 50\%, 75\%\}$ sparsity. The experiment evaluates performance with equal input and output lengths ($In_t$) and with output 10 times the input length ($Out_t$).

▷ **Zero-shot Experiments.** Specifically, we follow the setup from [45] and conduct two experiments. In the Prometheus, we train the model In-Domain environments $b_1, b_2, \ldots, b_{20}$ and evaluate its generalization ability in new environments $b_{11}, b_{12}$, using MSE as the evaluation metric. In the

Table 3: Comparison of Spatial & Temporal Generalization in the Prometheus benchmark.

| SPARSITY TRAIN ↓ | TEST→ | $s_{TS}=5\%$ | | $s_{TS}=25\%$ | | $s_{TS}=50\%$ | | $s_{TS}=75\%$ | |
|---|---|---|---|---|---|---|---|---|---|
| | | IN-T | OUT-T | IN-T | OUT-T | IN-T | OUT-T | IN-T | OUT-T |
| $s_{TR}=75\%$ | U-NET | 0.1847 | 0.2103 | 0.2345 | 0.2877 | 0.2654 | 0.3018 | 0.2273 | 0.3391 |
| | + PURE | 0.1622 | 0.1854 | 0.2079 | 0.2581 | 0.2365 | 0.2710 | 0.1998 | 0.3024 |
| | FNO | 0.0659 | 0.0872 | 0.0921 | 0.1232 | 0.1109 | 0.1821 | 0.2109 | 0.2455 |
| | + PURE | 0.0504 | 0.0654 | 0.0689 | 0.0946 | 0.0805 | 0.1417 | 0.1582 | 0.1883 |

| | Temperature Field | 75% | Smoke Field |
|---|---|---|---|
| Sparse Input | | | |
| Ground Truth | | | |
| Ours+PURE Error | | | |
| DGODE Error | | | |
| FNO Error | | | |
| U-Net Error | | | |

Figure 2: The top row shows the sparse input data used for predictions. The second row displays the true data for both fields. Red boxes highlight areas of significant error.

Navier-Stokes equations, we train the model on a $64 \times 64 \times 20$ dataset and evaluate it on a higher resolution $512 \times 512 \times 20$ dataset, focusing on the fluid dynamics details in the last five time steps and the handling of complex flow patterns and boundary layers.

## 4.2 Generalization Experiment Results

In this section, we focus on the issue of generalization. Based on our experimental findings, we make the following observations. **Out-of-Distribution Generalization.** The results as shown in Table 1. On the Prometheus dataset, PURE outperforms all benchmark models with an MSE of 0.0323 in-distribution and 0.0328 OOD. It improves over the second-best model, DGODE (MSE 0.0344 in-distribution, 0.0359 OOD), by 6.10% and 8.63%, respectively. On the Navier-Stokes dataset, PURE achieves the best performance with an MSE of 0.0752 in-distribution and 0.0763 OOD, improving by 6.58% and 26.07% over the second-best model. On the Spherical-SWE dataset, PURE has an MSE of 0.0022 both in-distribution and OOD, which is 41.46% better than the second-best model. Additionally, in Table 2, the performance of various benchmark models significantly improves when using PURE, in summary, the PURE framework performs excellently in handling OOD fluid dynamics modeling.

**Spatial & Temporal Generalization.** Table 3 shows that PURE excels in the Prometheus benchmark, notably reducing errors with sparse data. For instance, in the 75% sparsity test, U-Net's MSE drops from 0.2273 to 0.1998, and FNO from 0.2109 to 0.1582. Figure 2 highlights PURE's lower errors in temperature and smoke fields compared to DGODE, FNO, and U-Net, especially in red-boxed areas, showcasing its advantage in capturing complex dynamics. Additionally, PURE performs consistently across different prediction lengths; for example, U-Net's MSE decreases from 0.1847 to 0.1622 for in-time prediction (In-t) and from 0.2103 to 0.1854 for out-of-time prediction (Out-t). Overall, PURE excels with sparse and out-of-distribution data and enhances performance across prediction lengths, demonstrating strong spatial and temporal generalization.

**Visualization and Analysis.** Figure 3 compares the performance of different methods in fluid dynamics modeling, including the Prometheus dataset, Navier-Stokes equations, and the 3D Reaction-Diffusion Equation. In the Prometheus dataset, using PURE significantly reduces DGODE's prediction error, especially in complex dynamic regions. For the Navier-Stokes and Spherical Shallow Water equations, FNO and NMO models combined with PURE excel in capturing complex flow features. In the 3D Reaction-Diffusion Equation, DGODE with PURE significantly reduces prediction errors. Overall, PURE greatly enhances the prediction accuracy of models in fluid dynamics, allowing for better capture of complex dynamic evolution.

## 4.3 Zero-shot Super-resolution and Environment Generalization

As shown in Figure 4, the PURE framework performs excellently in zero-shot super-resolution and environmental generalization experiments. In the Prometheus benchmark, the FNO model using

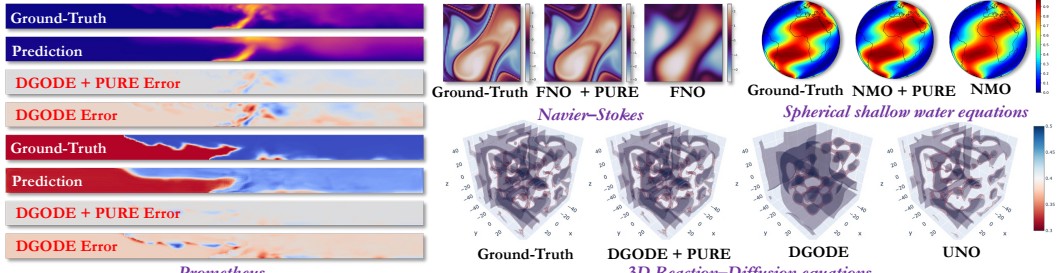

Figure 3: The Figure compares the performance of various methods in fluid dynamics modeling, including Prometheus, Navier-Stokes equations, and 3D reaction-diffusion equations. Models with PURE significantly reduce prediction errors in fluid dynamics, capturing complex dynamic evolutions.

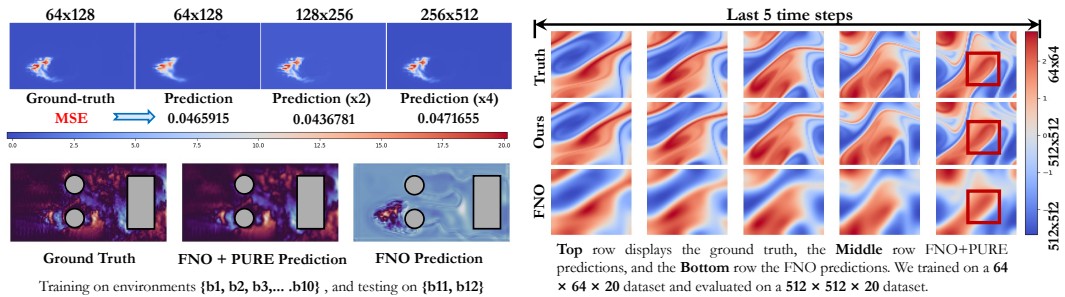

Figure 4: *Left*. Zero-shot super-resolution and environment generalization experiments on Prometheus. *Right*. Zero-shot super-resolution experiments on the Navier-Stokes equations.

PURE significantly reduces prediction errors at different resolutions, with an MSE of 0.0471655 at a $256 \times 512$ resolution. For the Navier-Stokes equations, the FNO model combined with PURE significantly reduces prediction errors on high-resolution datasets and performs better in handling complex flow patterns and boundary layers, especially in capturing details in the last five time steps. Overall, PURE significantly improves model prediction accuracy and generalization ability in zero-shot super-resolution and environmental generalization tasks.

## 4.4 Qualitative Analysis & Ablation Study

In this section, we evaluate the effectiveness of the PURE method and the importance of its components through qualitative analysis and ablation studies.

**Qualitative Analysis.** The Figure 5 uses t-SNE to perform clustering analysis on FNO prediction results. (a) represents the ground truth, (b) shows the predictions of the original FNO, and (c) shows the predictions of FNO combined with PURE. It is evident that the FNO combined with PURE is closer to the labels in clustering effect, with a more tightly distributed data point cluster. This demonstrates that PURE significantly improves the prediction accuracy of the FNO model.

**Ablation Study.** To evaluate the contribution and importance of each component in the proposed PURE, we design ablation experiments based on the default backbone model in this paper, and we use Relative L2 error as metric. Our model variants are as follows: **(1) PURE w/o Graph ODE**, we

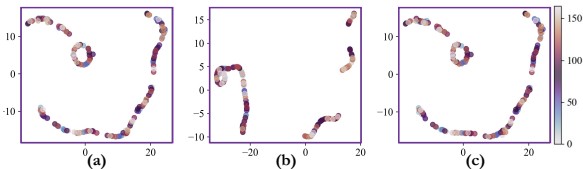

Figure 5: t-SNE clustering. (a) Ground truth, (b) FNO predictions, (c) FNO +PURE predictions.

Table 4: Ablation Studies on S-SWE.

| VARIANTS | S-SWE |
|---|---|
| PURE W/O GRAPH ODE | 0.1882 |
| PURE W/O INTERPOLATION | 0.1696 |
| PURE W/O MI | 0.1588 |
| PURE W/O FFT | 0.1602 |
| **PURE** | **0.1357** |

remove the Graph ODE module and use static prompt embeddings. **(2) PURE w/o Interpolation**, we remove interpolation and use only Eqn. 7. **(3) PURE w/o MI**, we remove the mutual information minimization. **(4) PURE w/o FFT**, we remove the frequency domain enhancement (FFT). Table 4 shows the results of our ablation study. Removing Graph ODE, interpolation, mutual information minimization, and FFT results in Relative L2 errors of 0.1882, 0.1696, 0.1588, and 0.1602, respectively. The complete PURE method has an error of 0.1357. The results of the ablation experiments show that removing any component results in a decrease in predictive performance, further proving the critical role of these components in the PURE method. More results in Appendix H.

## 5 Conclusion

In this paper, we study a practical problem of out-of-distribution fluid dynamics modeling and propose a novel approach named PURE for this problem. The high-level idea of our PURE is to learn time-evolving prompts using graph ODEs, which can effectively adapt spatio-temporal forecasting models to different scenarios. Our PURE first initializes prompt embeddings by exploring multi-view context information from spatio-temporal data and system parameters. Then, PURE incorporates the interpolation of observation sequences into the graph ODE, which helps capture the temporal evolution of prompt embeddings to mitigate temporal distribution shifts. In future works, we will extend our PURE to more real-world scenarios such as rigid dynamics modeling and traffic flow forecasting.

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

# A  Proofs of Theorem 3.1 and Theorem 3.2

*Proof of Theorem 3.1.* We first introduce a lemma as follows.

**Lemma A.1.** *Suppose that* $\dot{x} = M_1 x + b(t), x(0) = x_0$. *Then, the solution can be written as*

$$x(t) = x_0 e^{M_1 t} + \int_0^t b(s) e^{M_1(t-s)} ds. \tag{25}$$

The proof of Lemma A.1 is from Method of Variation of Parameters [37]. With Lemma A.1, we can prove Theorem 3.1.

By Lemma A.1, we have

$$x(t) = x_0 e^{M_1 t} + \int_0^t b(s) e^{M_1(t-s)} ds, \tag{26}$$

and

$$y(t) = x_0 e^{M_1 t} + \int_0^t b'(s) e^{M_1(t-s)} ds. \tag{27}$$

It follows that

$$x(t) - y(t) = \int_0^t (b(s) - b'(s)) e^{M_1(t-s)} ds. \tag{28}$$

By the Mean Value Theorem for Integrals, we have

$$x(t) - y(t) = (b(s') - b'(s')) \int_0^t e^{M_1(t-s)} ds. \tag{29}$$

Thus, we obtain that

$$|x(t) - y(t)| \geq c_0 (e^{M_1 t} - 1)/M_1, \tag{30}$$

which we complete the proof. $\qquad\square$

*Proof of Theorem 3.2.* To prove the theorem, we need the following lemma, which can be found in [5].

**Lemma A.2.** *Given the following ODEs in* $\mathbb{R}^n$,

$$\begin{aligned} \dot{x} &= A(t)x + F(x,t), & x(0) &= x_0, \\ \dot{y} &= A(t)y + G(y,t), & y(0) &= x_0, \end{aligned} \tag{31}$$

*assume that* $F$ *is globally Lipschitz continuous and "close to* $G$. *In other words, there exist* $L \geq 0$ *and* $\epsilon \geq 0$ *such that*

$$\begin{aligned} \|F(x,t) - F(y,t)\| &\leq L : \|x - y\|, & \text{for all } x, y \in \mathbb{R}^n \text{ and } t \in [0,T), \\ \|F(x,t) - G(x,t)\| &\leq \epsilon, & \text{for all } x \in \mathbb{R}^n \text{ and } t \in [0,T), \end{aligned} \tag{32}$$

*and assume that*

$$\|\Phi(t,s)\|_i \leq c e^{\eta(t-s)}, \quad \text{for all } 0 \leq s \leq t < T, \tag{33}$$

*in which* $\|\cdot\|_i$ *denotes the induced matrix norm associated with vector norm* $\|\cdot\|$, $\Phi(t,s)$ *denotes the transition matrix for* $A(t)$, *and* $c \geq 1$. *Then for all* $t \in [0,T)$, *if* $\eta + cL \neq 0$

$$\|x(t) - y(t)\| \leq \frac{\epsilon c}{\eta + cL} \left( e^{(\eta + cL)t} - 1 \right). \tag{34}$$

In this case, we set $c = 1, \eta = 0$. Then, Theorem 3.2 is clear from the Lemma A.2. $\qquad\square$

## B  Our Basic Forecasting Mode Details

Our base forecasting model combines two parallel modules [81]: the Fourier Neural Operator (FNO) [45] and a Vision Transformer (ViT)-based convolution [10]. The FNO processes the input observation embeddings in the frequency domain with Fast Fourier Transform (FFT) and inverse FFT (iFFT), capturing frequency features. The ViT module uses a multi-head attention mechanism to process the spatial features of the input data. The PURE framework generates time-evolving prompt embeddings using a Graph ODE, capturing dynamic changes in spatio-temporal features. These embeddings, along with the features from the FNO and ViT modules, are integrated using skip connections and a Multi-Layer Perceptron (MLP) to produce the final predictions. The model optimizes by reducing the mean squared error (MSE) between predictions and the ground truth, and by enhancing robustness through reducing mutual information between prompt and observation embeddings. This approach ensures high accuracy in out-of-distribution fluid dynamics forecasting.

## C  Related work

**Dynamical System Modeling.** The field combining machine learning with dynamical systems aims to use machine learning methods to model, predict, and control the behavior of dynamical systems [45, 80, 59, 55, 2, 54, 81]. Key techniques include using neural networks to extract patterns from spatiotemporal data, such as Convolutional Neural Networks (CNN) [63, 61], Graph Neural Networks (GNN) [59, 43, 38], and Transformer models [79, 4, 36]. Additionally, Physics-Informed Neural Networks (PINN) [34, 60] embed physical laws into neural networks to enhance the model's physical consistency. These methods apply to both short-term and long-term predictions of dynamical systems and optimize control strategies in areas like robotics and autonomous driving [11, 57]. To address the challenge of out-of-distribution data, researchers develop new datasets and benchmarks to evaluate model performance under different data distributions [80]. This field finds wide applications in aerospace, biomedical, and meteorological domains [93, 39]. In this work, we propose a framework named PURE, which uses prompt learning and graph neural ODE to address complex distribution shifts in fluid dynamics due to parameter and temporal changes.

**Out-of-distribution Generalization** Out-of-distribution (OOD) [71, 73, 83, 85, 27] generalization means a model performs well on new, unseen data. The core goal in this field is to improve model performance when training and test data come from different distributions. Models that excel in OOD scenarios should be robust and adaptable. Researchers have proposed several methods, such as data augmentation [74, 8], invariant feature learning [42, 77, 41, 72], adversarial training [75, 8], and domain adaptation [35, 15]. These methods are widely used in areas like autonomous vehicles, medical diagnosis, financial forecasting, and dynamical systems modeling [80, 13, 53]. In this work, we propose PURE, which uses prompt learning and graph neural ODE to adapt spatio-temporal forecasting models to address distribution shifts in fluid dynamics.

**Prompt Learning.** Prompt learning [48, 16, 24] has recently gained significant attention as a strategy for adapting pre-trained models to various downstream tasks by leveraging the power of prompt-based fine-tuning [9, 29, 94, 76, 52]. In the domain of large language models, prompt learning aims to incorporate optimal tokens into the input sequence, which can effectively improve performance without extensive retraining [28, 92, 90]. In the context of fluid dynamics modeling, prompt learning means a supplementary hint to indicate the context, which is incorporated into the input (observation embedding) for better generalization. Although it shares a similar meaning as prompt tuning in language models, our prompt refers to the current environment, which determines the future evolution with better generalization.

## D  The Proposed PURE Algorithm

The whole learning algorithm of PURE is summarized in Algorithm 1.

## E  Detailed description of datasets

We evaluate our proposed  PURE on five physical benchmarks.

---

**Algorithm 1** PURE Framework

---

**Require:** Historical observations $\{s_i^{1:T_0}\}_{i=1}^N$, physical parameters $\boldsymbol{\xi}$
**Ensure:** Future observations $\{s_i^{T_0+1:T_0+T}\}_{i=1}^N$
 1: **Initialize** prompt embeddings using Multi-view Context Exploration
 2: **for** each sensor $i$ **do**
 3:     Map location $\boldsymbol{x}_i$ and initial observation $\boldsymbol{s}_i^0$ to embeddings $\boldsymbol{p}_i$ and $\boldsymbol{q}_i$ using FFNs $\phi^{PE}(\cdot)$ and $\phi^{OE}(\cdot)$
 4:     Aggregate embeddings: $\boldsymbol{e}_i = \boldsymbol{p}_i \odot \boldsymbol{q}_i$
 5: **end for**
 6: Stack initial embeddings $\boldsymbol{E}^0 = \{\boldsymbol{e}_i\}_{i=1}^N$ and apply self-attention blocks $\phi^{SA,(l)}$
 7: Retrieve representations $\boldsymbol{u}^q$ for each query position $\boldsymbol{x}_q$ using attention mechanism
 8: Generate 3D representation tensor $\boldsymbol{U}$ and integrate with parameter embedding $\boldsymbol{u}^p = \phi^{PA}(\boldsymbol{\xi})$ to obtain $\tilde{\boldsymbol{U}}$
 9: Enhance representation in frequency domain: $\boldsymbol{H} = \text{iFFT}(\text{FFN}(\text{FFT}(\tilde{\boldsymbol{U}})))$
10: Flatten tensor $\boldsymbol{H}$ and retrieve initial prompt embeddings $\boldsymbol{z}_i^0$
11: **Learn Time-evolving Prompts with Graph ODE**
12: **for** each timestamp $t$ **do**
13:     Interpolate observations $\boldsymbol{s}_i^t$ from historical sequences
14:     Update prompt embeddings $\boldsymbol{z}_i^t$ using continuous graph ODE
15:     Incorporate interpolated observations into graph ODE with attention mechanism
16: **end for**
17: **Model Adaptation with Prompt Embeddings**
18: **for** each sensor $i$ **do**
19:     Concatenate observation embeddings $\boldsymbol{\mu}_i^t$ and prompt embeddings $\boldsymbol{z}_i^t$ to obtain $\tilde{\boldsymbol{\mu}}_i^t$
20: **end for**
21: Incorporate $\tilde{\boldsymbol{\mu}}_i^t$ into spatio-temporal forecasting model
22: Optimize framework by minimizing MSE loss $\mathcal{L}_{MSE}$ and mutual information loss $\mathcal{L}_{MI}$
23: **return** Predicted future observations $\{s_i^{T_0+1:T_0+T}\}_{i=1}^N$ =0

---

*Prometheus* [80] is a large-scale fluid dynamics dataset focused on studying out-of-distribution (OOD) generalization. This dataset simulates tunnel and pool fire scenarios, generating 4.8TB of raw data compressed to 340GB. The tunnel fire simulation takes place in a tunnel 100 meters long, 6 meters wide, and 6 meters high. It adjusts the heat release rate (HRR) and ventilation speed to create 30 different environmental combinations. The pool fire simulation occurs in a 150x100 meter area with tanks and buildings, creating 25 different environmental combinations by adjusting HRR and ventilation speed. Each scenario includes a high-density sensor network to measure temperature and gas concentration. The dataset integrates advanced engineering methods, focusing on precise and efficient data analysis and inference on irregular grid structures. Prometheus provides rich data resources and benchmarks for OOD generalization research in fluid dynamics.

*Navier-Stokes equations* [45] depict the motion of a viscous, incompressible fluid. The equations are as follows in vorticity form on the unit torus:

$$
\begin{aligned}
\partial_t w(x,t) + u(x,t) \cdot \nabla w(x,t) &= \nu \Delta w(x,t) + f(x), \quad x \in (0,1)^2, \quad t \in (0,T] \\
\nabla \cdot u(x,t) &= 0, \quad x \in (0,1)^2, \quad t \in [0,T] \\
w(x,0) &= w_0(x), \quad x \in (0,1)^2
\end{aligned}
\tag{35}
$$

We solve these equations using the stream function formulation and a pseudo-spectral approach. In particular, we first solve the Poisson equation in order to identify the velocity field. Afterward, we differentiate vorticity, compute the nonlinear terms, and apply de-aliasing. We use the Crank-Nicolson scheme for time-stepping, recording the solution at time intervals of $t = 1$ on a 256×256 grid followed by downsampling. For the Bayesian inverse problem, the timestep during data generation is 1e-4, and in MCMC, it is 2e-2. We simulate the Navier-Stokes equations with varying viscosity coefficients, adjusting $\nu$ to study its impact on fluid flow and vorticity distribution.

*Spherical Shallow Water Equations (Spherical-SWE)* [14] describe the large-scale atmospheric and oceanic fluid motion on Earth's surface. The equations are:

$$\partial_t h + \nabla \cdot (h\mathbf{u}) = 0$$
$$\partial_t \mathbf{u} + (\mathbf{u} \cdot \nabla)\mathbf{u} + f\mathbf{k} \times \mathbf{u} = -g\nabla h + \nu\Delta\mathbf{u} \tag{36}$$

Here, $h$ represents fluid thickness, $\mathbf{u}$ is the fluid velocity vector, $f$ represents the Coriolis parameter, $\mathbf{k}$ represents the unit vertical vector, $g$ represents the gravitational acceleration, $\nu$ represents the viscosity coefficient, and $\Delta$ represents the Laplacian operator. The first equation (continuity equation) represents mass conservation, describing changes in fluid thickness. The second equation (momentum equation) represents momentum conservation, including advection, Coriolis force, pressure gradient force, and viscous diffusion. We simulate the Spherical Shallow Water Equations with different viscosity coefficients $\nu$, adjusting $\nu$ to study its effects on fluid motion.

*3D Reaction-Diffusion Equations* [62] describe the diffusion and reaction of chemical substances in space. The general form of these equations is:

$$\partial_t u = D_u \Delta u + R_u(u, v)$$
$$\partial_t v = D_v \Delta v + R_v(u, v) \tag{37}$$

Here, $u$ and $v$ represent the concentrations of the chemical substances, $D_u$ and $D_v$ represent the diffusion coefficients for $u$ and $v$, respectively, $\Delta$ is the Laplacian operator, and $R_u(u, v)$ and $R_v(u, v)$ denote the reaction terms that represent the reaction rates between $u$ and $v$. Diffusion terms, i.e., $\Delta u$ and $\Delta v$ denote the diffusion process of the chemicals in space. The diffusion coefficients $D_u$ and $D_v$ determine the rate of diffusion. Reaction terms, i.e., $R_u(u, v)$ and $R_v(u, v)$ describe the reaction rates of the chemical substances. These terms depend on the concentrations of $u$ and $v$, and can include linear reactions, nonlinear reactions, and complex dynamic processes. We simulate the 3D reaction-diffusion equations with different diffusion coefficients $D_u$ and $D_v$ to study the effects of diffusion rates on the distribution and reaction rates of the chemical substances.

*ERA5* [23] is a global atmospheric reanalysis dataset produced by ECMWF, providing weather data from 1979 to the present with high spatial (31 km) and temporal (hourly) resolution. It includes variables like surface pressure, sea surface temperature, sea surface height, and two-meter temperature. ERA5 data supports applications in weather forecasting, climate research, environmental monitoring, energy management, and agriculture. Accessible via the Copernicus Climate Data Store, ERA5 is crucial for analyzing and predicting meteorological and climate phenomena.

Table 5: The table presents the *In-Domain* and *Adaptation* environments for various benchmarks. Training and testing in the *In-Domain* environment is called *w/o* OOD experiment, while training in the *In-Domain* environment and testing in the *Adaptation* environment is called *w/* OOD experiment.

| BENCHMARKS | IN-DOMAIN ENVIRONMENTS | ADAPTATION ENVIRONMENTS |
|---|---|---|
| PROMETHEUS | $\{a_1, a_2, ..., a_{25}\}, \{b_1, b_2, ..., b_{20}\}$ | $\{a_{26}, a_{27}, ..., a_{30}\}, \{b_{21}, b_{22}, ..., b_{25}\}$ |
| 2D NAVIER-STOKES EQUATION | $\nu = \{1e^{-1}, 1e^{-2}, ..., 1e^{-9}, 1e^{-10}\}$ | $\nu = \{1e^{-11}, 1e^{-12}\}$ |
| SPHERICAL SHALLOW WATER EQUATION | $\nu = \{1e^{-1}, 1e^{-2}, ..., 1e^{-9}, 1e^{-10}\}$ | $\nu_t = \{1e^{-11}, 1e^{-12}\}$ |
| 3D REACTION–DIFFUSION EQUATIONS | $D = \{2.1 \times 10^{-5}, 1.6 \times 10^{-5}, 6.1 \times 10^{-5}\}$ | $D = \{2.03 \times 10^{-9}, 1.96 \times 10^{-9}\}$ |
| ERA5 | $V = \{Sp, SST, SSH, T2m\}$ | $V = \{SSR, SSS\}$ |

## F  Details of Compared Approaches

The compared approaches involved in this study is as follows:

- U-Net [64] is a convolutional neural network initially used for biomedical image segmentation. It has a symmetric U-shaped structure and uses skip connections to link the encoder and decoder, enabling efficient feature fusion.

- ResNet [21] introduces residual blocks to solve the degradation problem in deep networks. It allows the network to be deeper and easier to train by using skip connections to directly pass information.

- ViT [10] applies the Transformer model to image recognition. It divides the image sample into patches and uses self-attention mechanisms to process these patches, balancing computational efficiency and performance.

- SwinT [49] introduces a sliding window mechanism for effective local and global feature extraction. It is suitable for various computer vision tasks.
- FNO [45] uses Fourier transforms for global feature extraction, suitable for processing continuous field data and efficiently solving PDEs.
- UNO [1] combines the U-Net architecture with optimization methods to enhance feature extraction and fusion capabilities, improving model performance.
- CNO [63] combines convolution operations with operator learning, focusing on high-dimensional continuous data and modeling complex dynamic systems.
- NMO [82] enhances the modeling capability for multi-scale dynamic systems by combining neural networks with manifold learning algorithms.
- CGODE [26] is a neural ODE model that aims to capture the dynamics of both nodes and edges jointly.
- DGODE [80] addresses the challenge of out-of-distribution (OOD) generalization in fluid dynamics modeling by learning disentangled representations using a temporal GNN and a frequency network. It minimizes mutual information between node and environment representations to mitigate distribution shifts and employs a coupled graph ODE framework for robust modeling.

## G Metrics details

**Mean Squared Error (MSE).** Mean Squared Error measures the gap between the predicted and ground truth. The formula is:

$$\text{MSE} = \frac{1}{n} \sum_{i=1}^{n} (y_i - \hat{y}_i)^2, \tag{38}$$

in which $y_i$ denotes the actual value, $\hat{y}_i$ denotes the predicted value, and $n$ denotes the number of data points.

**Relative L2 Error.** Relative L2 Error evaluates the relative accuracy of the model's predictions. The formula is:

$$\text{Relative L2 Error} = \frac{\|\mathbf{y} - \hat{\mathbf{y}}\|_2}{\|\mathbf{y}\|_2}, \tag{39}$$

where $\|\cdot\|_2$ denotes the L2 norm, $\mathbf{y}$ is the vector of actual values, and $\hat{\mathbf{y}}$ is predicted values.

## H More experiment results

▷ **Sparse Reconstruction Experiments.** The experimental setup includes sparse reconstruction experiments on the Prometheus and ERA5 datasets. For the Prometheus dataset, the sparsity rates are set to 25%, 50%, and 75%. For the ERA5 dataset, the sparsity rates are set to 5% and 25%. Each set of experimental results includes the original data, sparse input data, results from our method, and results from MMGNet [56]. The experiments evaluate the performance of each method by comparing the reconstruction results at different sparsity rates.

Figure 6 shows two sets of sparse reconstruction experiment results from the Prometheus and ERA5 datasets. The sparsity rates for Prometheus are 25%, 50%, and 75%, while for ERA5, they are 5% and 25%. Each set displays the Ground Truth, Sparse Input, our reconstruction method (Ours), and MMGNet's results in order. As the sparsity rate increases, the quality of the reconstruction decreases. At low sparsity rates, both our method and MMGNet effectively restore image details. At high sparsity rates, our method performs better in preserving details and recovering overall structure.

▷ **More Ablation Experiments.** Table 6 show the ablation study outcomes on the Navier-Stokes equations. We use MSE to assess the contribution of each component. We remove different components from the PURE method and compare them with the original FNO model and the complete FNO + PURE method. The complete FNO + PURE method achieves the lowest MSE of 0.0987, while the original FNO model has an error of 0.1567. Removing Graph ODE, interpolation, mutual information minimization, and FFT increases the error to 0.1282, 0.1097, 0.1182, and 0.1266, respectively. These results clearly show that each component of the PURE method significantly improves

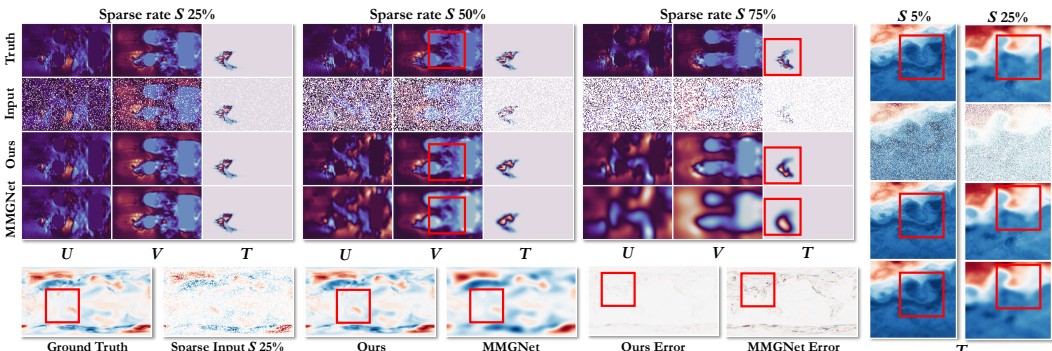

Figure 6: The figure shows sparse reconstruction results using the Prometheus and ERA5 datasets at various sparsity rates. Each group displays the ground truth, sparse input, our reconstruction method, and MMGNet's results. As the sparsity rate increases, the quality of the reconstruction decreases. U and V represent velocity components, and T represents temperature.

Table 6: Ablation Studies on Navier-Stokes equations (with a viscosity coefficient of $\nu = \{1e^{-3}\}$.

| VARIANTS | NAVIER-STOKES EQUATIONS |
|---|---|
| FNO + PURE W/O GRAPH ODE | 0.1282 |
| FNO + PURE W/O INTERPOLATION | 0.1097 |
| FNO + PURE W/O MI | 0.1182 |
| FNO + PURE W/O FFT | 0.1266 |
| FNO | 0.1567 |
| FNO + **PURE** | **0.0987** |

the model's predictive performance. Removing any component leads to performance degradation, proving the importance of these components in enhancing prediction accuracy.

▷ **Performance with respect to Different Difficulty Levels.** Here, we demonstrate the performance of our PURE with varying difficulty levels. In particular, we measure the difficulty levels based on the distance between $P_{\text{train}}(\xi)$ and $P_{\text{test}}(\xi)$ and generate three levels on the Prometheus dataset. The compared results are shown in Table 7. From the results, we can observe that all the model performs worse in hard scenarios and our method has consistently outperformed these baselines. The potential reason is that (1) our model enhances model invariance across different distributions through decoupling prompt embeddings and observation embeddings via mutual information which results in high generalizationability to different environments; (2) our model utilizes multi-view context mining and graph ODE to extract prompt embeddings, which capture environment information accurately.

Table 7: Performance comparison across varying levels of OOD generalization difficulty. The values represent the Mean Squared Error (MSE) for each method.

| METHOD | U-NET | RESNET | VIT | SWIN-T | FNO | CGODE | PURE |
|---|---|---|---|---|---|---|---|
| EASY | 0.0945 | 0.0682 | 0.0654 | 0.0676 | 0.0452 | 0.0772 | **0.0325** |
| MID | 0.1063 | 0.0922 | 0.0902 | 0.0912 | 0.0544 | 0.0863 | **0.0341** |
| HARD | 0.1432 | 0.1234 | 0.1076 | 0.1123 | 0.0623 | 0.0921 | **0.0354** |

▷ **Robustness to Noisy Data.** We have also experimented with noisy data to evaluate the robustness of our method. The results are shown in Table 8. From the results, the performance of both ResNet and NMO models degrades significantly when noise is introduced. However, the integration of our PURE with these models substantially mitigates the impact of noise, leading to much lower MSE values compared to their baselines.

▷ **Expanded Evaluation of OOD Generalization in Dynamical Systems**. To further validate the effectiveness of our PURE, we include three additional models specifically designed for out-of-distribution generalization in dynamical systems: LEADS [87], CODA [31], and NUWA [74]. Table 9 shows the performance of each method across different datasets in both in-distribution (ID) and out-of-distribution (OOD) scenarios. The results indicate that our proposed method outperforms these baselines on all datasets, especially in OOD scenarios.

Table 8: Performance comparison under noisy data conditions. The values represent the Mean Squared Error (MSE) for each method with and without noise.

| Dataset | ResNet/Noise | ResNet+PURE/Noise | NMO/Noise | NMO+PURE/Noise |
|---|---|---|---|---|
| **PROMETHEUS** | 0.0674 / 0.3422 | 0.0542 / 0.0586 | 0.0397 / 0.1287 | **0.0281 / 0.0309** |
| **NS** | 0.1823 / 0.6572 | 0.1492 / 0.1537 | 0.1021 / 0.2542 | **0.0876 / 0.0892** |

Table 9: Performance comparison of our method (PURE) against additional baselines on various datasets. The values represent Mean Squared Error (MSE) in in-distribution (ID) and out-of-distribution (OOD) scenarios.

| Dataset | Prometheus | | ERA5 | | SSWE | |
|---|---|---|---|---|---|---|
| | ID | OOD | ID | OOD | ID | OOD |
| **LEADS** | 0.0374 | 0.0403 | 0.2367 | 0.4233 | 0.0038 | 0.0047 |
| **CODA** | 0.0353 | 0.0372 | 0.1233 | 0.2367 | 0.0034 | 0.0043 |
| **NUWA** | 0.0359 | 0.0398 | 0.0645 | 0.0987 | 0.0032 | 0.0039 |
| **PURE (Ours)** | **0.0323** | **0.0328** | **0.0398** | **0.0401** | **0.0022** | **0.0024** |

# I   Limitations of This Study

Although the PURE method shows superiority on multiple benchmark datasets, it has some limitations. First, we assume that training and test data are independent and identically distributed (IID). This may be not true in some extreme physical scenarios due to environmental changes causing significant data distribution shifts. Second, while the PURE method performs well in addressing distribution shifts, it may still face challenges when dealing with high-dimensional and complex fluid dynamics systems. Additionally, the PURE method has high computational complexity, requiring more computational resources and time in practical applications. Future research can focus on optimizing the algorithm to improve computational efficiency and extending it to more real-world scenarios such as rigid dynamics modeling and traffic flow forecasting.

# J   Borader Impact

The PURE method significantly impacts out-of-distribution fluid dynamics modeling. It adapts to different scenarios, improving the model's performance in handling distribution changes. This is helpful for climate prediction, epidemic spread, aerospace, and biomedical fields. In future works, we will extend our PURE to more real-world scenarios such as rigid dynamics modeling and traffic flow forecasting.

