# OpenReview forum: "PURE: Prompt Evolution with Graph ODE for Out-of-distribution Fluid Dynamics Modeling"
_NeurIPS.cc/2024/Conference — NeurIPS 2024 poster_

### Official Review · Reviewer_MFD5 · 2024-07-08

**Soundness:** 3
**Presentation:** 2
**Contribution:** 2
**Rating:** 5
**Confidence:** 4

**Summary:**

The paper presents a new approach called Prompt Evolution with Graph ODE (PURE) for non-distributed fluid dynamics modeling. PURE first learns from historical observations and system parameters in the frequency domain to explore multi-view contextual information, which can efficiently initialize the cue embedding. Interpolations of the observation sequences are then merged into the graph ODE so that the time evolution of the model-adaptive cue embeddings can be captured. These time-evolving cue embeddings are then incorporated into the underlying predictive model to overcome spatio-temporal distributional variations. In addition the paper minimizes the mutual information between the cue embeddings and the observation embeddings to enhance the robustness of the model to different distributions. Finally, extensive experiments conducted on various kinds of benchmark datasets validate the superiority of the proposed PURE compared to various baselines.

**Strengths:**

1. The idea of the paper is novel. It is the first to link prompt learning to dynamic system modeling for out-of-distribution problems.

2. This paper is technically sound. PURE first learns initialized prompt embeddings from historical observations and system parameters, and then employs a graph ODE with interpolated observation sequences to capture the continuous evolution of their model adaptation under out-of-distribution changes.

3. The experimental results show the effectiveness of PURE in different challenging environments.

**Weaknesses:**

1. The contribution of the proposed method in dealing with the OOD problem needs to be further clarified since the advantages of PURE over the previous efforts, such as Refs. [7, 67, 14, 72], etc., to address the OOD problem are not listed.

2. The writing of the paper needs to be improved. Some of the symbols in the method description section are not defined, e.g., what do P and N in Equation 9 refer to?

3. The experiment is not comprehensive enough. (a) The reasons for selecting baselines are not explained. Data augmentation [66, 7], invariant feature learning [39, 69, 38], adversarial training [67, 7], and domain adaptation [32, 14] are mentioned in the paper in related work for solving the OOD problem, but they are not be compared as baselines in the experiment. (b) The experiments in this paper do not state whether noisy data are considered. (c) The authors just give a brief description of the results without analyzing the reasons behind the high performance.

**Questions:**

1.  What are the advantages of PURE over previous efforts to address OOD?

2. Some of the symbols in the method description section are not defined, e.g., what do P and N in Equation 9 refer to?

**Limitations:**

The authors list limitations in the appendix but do not mention them in the main text. It is recommended that the author make a description in the main text.

---

> ### Author Rebuttal · Authors · 2024-08-07
>
> We are truly grateful for the time you have taken to review our paper, your insightful comments and support. Your positive feedback is incredibly encouraging for us! In the following response, we would like to address your major concern and provide additional clarification.
>
> > Q1. The contribution of the proposed method in dealing with the OOD problem needs to be further clarified since the advantages of PURE over the previous efforts, such as Refs. [7, 67, 14, 72], etc., to address the OOD problem are not listed.
>
> A1. Thanks for your comment. Compared with previous OOD methods, our contribution can be listed as follows:
> - **Underexplored Scenarios**. Our work studies an underexplored yet practical problem of OOD generalization in fluid dynamic modeling while previous OOD methods usually study the OOD problem in image classification scenarios.
> - **Innovative Methodlogy**. Our work not only learns time-evolving prompts using context mining and a graph ODE but also decouples prompt embeddings and observation embeddings via mutual information for invariant learning to address the OOD problem.
> - **Theoretical Analysis**. We provide a comprehensive theoretical analysis to support our designs, which makes our framework more solid.
> - **Superior Performance.** Comprehensive experiments validate the effectiveness of our method in different challenging settings. In particular, it improves performance by an average of 25.90% in OOD scenarios.
>
> > Q2. The writing of the paper needs to be improved. Some of the symbols in the method description section are not defined, e.g., what do P and N in Equation 9 refer to?
>
> A2. Thank you for your comment. $P$ denotes the set collecting all the positive pairs (i.e., from the same trajectory) of observation embeddings and prompt embeddings while $N$ denotes the set collecting all the possible pairs of observation embeddings and prompt embeddings in the dataset. We will include it in our revised version.
>
>
> > Q3. The experiment is not comprehensive enough. (a) The reasons for selecting baselines are not explained. Data augmentation [66, 7], invariant feature learning [39, 69, 38], adversarial training [67, 7], and domain adaptation [32, 14] are mentioned in the paper in related work for solving the OOD problem, but they are not be compared as baselines in the experiment.
>
> A3. Thank you for your comment. We have added three more baselines LEADS [1], CODA [2] and NUWA [3] for performance comparison. The results show that our method is superior to these baselines, which validates the effectiveness of our method when addressing OOD problems. In addition, the other references focus on the image classification problem, which cannot be adopted to solve our fluid dynamics modeling problem. We will include it in our revised version.
>
> | Dataset     | Prometheus (ID) | Prometheus (OOD) | ERA5 (ID) | ERA5 (OOD) | SSWE (ID) | SSWE (OOD) |
> |-------------|----------------|----------------|---------|----------|---------|----------|
> | LEADS       | 0.0374         | 0.0403         | 0.2367  | 0.4233   | 0.0038  | 0.0047   |
> | CODA        | 0.0353         | 0.0372         | 0.1233  | 0.2367   | 0.0034  | 0.0043   |
> | NUWA        | 0.0359         | 0.0398         | 0.0645  | 0.0987   | 0.0032  | 0.0039   |
> | PURE (ours) | 0.0323         | 0.0328         | 0.0398  | 0.0401   | 0.0022  | 0.0024   |
>
> > Q4. (b) The experiments in this paper do not state whether noisy data are considered.
>
> A4. Thank you for your comment. We have added experiments with noisy data to evaluate the robustness of our method. The results are shown below, which validate that our method is more robust to interference from noisy data. We will include it in our revised version.
>
> |            | ResNet/Noise | ResNet+PURE/Noise | NMO/Noise   | NMO+PURE/Noise |
> | ---------- | ------  | ----------------- | ------ | -------------- |
> | PROMETHEUS | 0.0674/0.3422 | 0.0542/0.0586     | 0.0397/0.1287 | 0.0281/0.0309  |
> | NS         | 0.1823/0.6572 | 0.1492/0.1537     | 0.1021/0.2542 | 0.0876/0.0892  |
>
>
>
> >Q5. (c) The authors just give a brief description of the results without analyzing the reasons behind the high performance.
>
> A5. Thank you for your comment. The potential reason for our high performance is that (1) our model enhances model invariance across different distributions through decoupling prompt embeddings and observation embeddings via mutual information which results in high generalizationability to different environments; (2) our model utilizes multi-view context mining and graph ODE to extract prompt embeddings, which capture environment information accurately. We will include it in our revised version.
>
> **Reference**
>
> [1] Kirchmeyer, Matthieu, et al. "Generalizing to new physical systems via context-informed dynamics model." ICML 2022.
>
> [2] Yin, Yuan, et al. "LEADS: Learning dynamical systems that generalize across environments." NeurIPS2021.
>
> [3] Wang, Kun, et al. "NuwaDynamics: Discovering and Updating in Causal Spatio-Temporal Modeling." ICLR2024.
>
> We will also add your suggestion about future works to our revised version. Thanks again for appreciating our work and for your constructive suggestions. Please let us know if you have further questions.

---

> ### Author Response · Authors · 2024-08-14
> **Summary of our rebuttal**
>
> Dear Reviewer,
>
> We summarized our rebuttal content at your convenience as follows:
> - We have included a detailed explanation of our contribution.
> - We have included more competing baselines to demonstrate the superiority of our approach.
> - We have included more real-world settings such as noisy scenarios.
> - We have included more analysis about our superiority.
>
> We will also add your suggestion about future works to our revised version. Thanks again for appreciating our work and for your constructive suggestions. Please let us know if you have further questions.
>
> Best,
>
> the Authors

---

### Official Review · Reviewer_vRkX · 2024-07-11

**Soundness:** 2
**Presentation:** 3
**Contribution:** 2
**Rating:** 4
**Confidence:** 3

**Summary:**

- The paper aims to improve the out-of-distribution (OOD) generalization of fluid dynamics modeling.

- Two types of OOD scenarios are targeted: OOD across different systems and OOD within the same system across different timestamps.

- The paper proposes a framework named PURE, composed of modules including:
    - Multi-view Context Exploration, which explores spatio-temporal data using both the attention mechanism and the frequency domain;
    - Time-evolving Prompt Learning, which incorporates the interpolation of observation sequences;
    - Model Adaptation with Prompt Embeddings, which leverages time-evolving prompts to mitigate temporal distribution shifts.

- Extensive experiments on a range of fluid dynamics datasets support the claim.

**Strengths:**

- Significant topic: OOD generalization in fluid dynamics modeling.

- Well-motivated, as OOD generalization is a crucial challenge in this field.

- The presentation effectively delivers the message.

- Extensive experiments have been conducted.

**Weaknesses:**

- My major concern with the paper is that the OOD challenge in dynamics modeling is not well-formulated. The paper describes the OOD scenario verbally as "*different dynamical systems could involve different parameters in underlying rules*" and "*during long-term auto-regressive forecasting, the input data distribution could vary hugely during temporal evolution,*" which is straightforward and easy to understand. However, the mathematical formulation of these scenarios is absent. This formulation should be the foundational basis of the topic, as we need to clearly define the problem before addressing it.

- Given the lack of mathematical formulation of the challenge, I find myself lost in the proposed approach section, unsure of the necessity for specific components. While I understand the function of each component, I cannot see why it is needed or which gaps it aims to bridge in the absent mathematical framework.

- Why is the proposed method termed "prompt"? Is there a connection to prompt tuning in large language models?

- How do you quantify the distribution shift in dynamics modeling? Can you rank the 'difficulty level' of OOD generalization in your experiments and analyze in which scenarios your method stands out and why?

**Questions:**

NA

---

> ### Author Rebuttal · Authors · 2024-08-07
>
> We are truly grateful for the time you have taken to review our paper and your insightful review. Here we address your comments in the following.
>
> >Q1. My major concern with the paper is that the OOD challenge in dynamics modeling is not well-formulated. The paper describes the OOD scenario verbally as "different dynamical systems could involve different parameters in underlying rules" and "during long-term auto-regressive forecasting, the input data distribution could vary hugely during temporal evolution," which is straightforward and easy to understand. However, the mathematical formulation of these scenarios is absent. This formulation should be the foundational basis of the topic, as we need to clearly define the problem before addressing it.
>
>
> A1. Thank you for your comment. In dynamical systems, the OOD problem studies the prediction performance of models under parameter distributions or environments not seen during training. In formulation, the evolution of dynamical systems is defined by $\frac{du}{dt} = F(u, \xi)$, where $u$ is the observation, and $\xi$ is the system parameter. If these parameters come from a distribution $\xi \sim P(\xi)$, the state trajectory comes from the distribution $u^{1:T_0} \sim P(u^{1:T_0} | \xi)$. Assume we learn a state mapping $f$ from time $u^{1:T_0}$ to $u^{T_0+1:T_0+T}$, i.e., $u^{T_0+1:T_0+T}= f(u^{1:T_0})$. We could have different distributions across training and test datasets, i.e., $P_{\text{train}}(\xi)\neq P_{\text{test}}(\xi)$, which results in $P_{\text{train}}(u^{1:T_0})\neq P_{\text{test}}(u^{1:T_0})$. Moreover, when conducting rollout prediction, we need to feed the output back to the model, i.e., $u^{T_{start}:T_{start}+T-1}= f(u^{T_{start}-T_0:T_{start}-1})$ with $P(u^{1:T_0} | \xi)\neq P(u^{T_{start}-T_0:T_{start}-1}|\xi, T_{start})$, which demonstrates temporal distribution shift. We will include our explanation in the revised version.
>
> > Q2. Given the lack of mathematical formulation of the challenge, I find myself lost in the proposed approach section, unsure of the necessity for specific components. While I understand the function of each component, I cannot see why it is needed or which gaps it aims to bridge in the absent mathematical framework.
>
> A2: Thank you for your comment. From our definition, the mapping $u_{output}=f(u_{input})$ could suffer from a serious distribution shift result from different $\xi$ and $T_{start}$, i.e., $P(u_{input}|\xi, T_{start})$. To reduce the impact of distribution shift, we aim to learn invariant observation embeddings $\mu^t$ to environments, i.e., $\xi$ and $T_{start}$, and utilize prompt embeddings $z^t$ to indicate the current environment. The basic idea of our method is to ensure the invariance of observation embeddings for better generalization, i.e., $z^t\perp \mu^t$. Then, the prompt embeddings would be combined to generate the future predictions, i.e., $u_{output} = \phi ([\mu^t,z^t])$. In our framework, a basic model is adopted to generate observation, i.e., $\mu^t=Basic Model(u_{input})$. We adopt context mining and graph ODE to learn time-varying prompt embeddings., i.e., $z^0= Context Mining (u_{input})$, $z^t = GraphODE (z^0,t)$, which can explore the temporal evolution of environments. We will include our explanation in the revised version.
>
> > Q3 Why is the proposed method termed "prompt"? Is there a connection to prompt tuning in large language models?
>
> A3. Thank you for your comment. "Prompt" means a supplementary hint to indicate the context, which is incorporated to the input (observation embedding) for better generalization. Generally, it shares the basic idea with prompt tuning, which also aims to incorporate optimal tokens into the input sequence. However, prompt tuning usually utilizes prompts to indicate different tasks in NLP scenarios, our prompt refers to the current environment in fluid dynamical modeling, which determines the future evolution with better generalization. We will include it in our revised version.
>
>
> > Q4. How do you quantify the distribution shift in dynamics modeling? Can you rank the 'difficulty level' of OOD generalization in your experiments and analyze in which scenarios your method stands out and why?
>
> A4. Thank you for your comment. We have added experiments to demonstrate the performance with varying difficulty levels. In particular, we measure the difficulty levels based on the distance between $P_{\text{train}}(\xi)$ and $P_{\text{test}}(\xi)$ and generate three levels on the Prometheus dataset. The compared results are shown below. We can observe that although all the model performs worse in hard scenarios, our method consistently outperforms these baselines. The potential reason is that (1) our model enhances model invariance across different distributions through decoupling prompt embeddings and observation embeddings via mutual information which results in high generalizationability to different environments; (2) our model utilizes multi-view context mining and graph ODE to extract prompt embeddings, which capture environment information accurately. We will include it in our revised version.
>
>
> | Method      | U-NET  | ResNet | VIT    | Swin-T | FNO    | CGODE  | PURE(Ours) |
> |-------------|--------|--------|--------|--------|--------|--------|------------|
> | **Easy**    | 0.0945 | 0.0682 | 0.0654 | 0.0676 | 0.0452 | 0.0772 | **0.0325**    |
> | **Mid**     | 0.1063 | 0.0922 | 0.0902 | 0.0912 | 0.0544 | 0.0863 | **0.0341**    |
> | **Hard**    | 0.1432 | 0.1234 | 0.1076 | 0.1123 | 0.0623 | 0.0921 |  **0.0354**     |
>
>
> **Reference**
>
> [1] Wu, et al. "Prometheus: Out-of-distribution Fluid Dynamics Modeling with Disentangled Graph ODE." ICML2024
>
> In light of these responses, we hope we have addressed your concerns, and hope you will consider raising your score. If there are any additional notable points of concern that we have not yet addressed, please do not hesitate to share them, and we will promptly attend to those points.

---

> > ### Comment · Reviewer_vRkX · 2024-08-10
> > **Thank you for your responses**
> >
> > Thank you for your responses. I appreciate the authors provide additional explanation to what I asked. I would like to maintain my rating (4) for the reason that
> > - I think the manuscript needs significant update to reflect the new information.
> > - The new information including the problem formulation (OOD in dynamic systems) is too fundamental rather trivial, which should appear at the beginning of the paper (problem setup).
> > - It is essential in motivating the proposal, including the design of different components, on how they can address the dynamic OOD (on what parameter invariance across training-test).
> > - Without revision, I fail to connect the problem to the proposal by looking the rebuttal text and the submission forth-and-back.

---

> ### Author Response · Authors · 2024-08-11
> **Thanks for your feedback!**
>
> Thanks for your feedback, and we have finished the revision based on your helpful suggestions.
>
> >Q1. The new information on OOD in dynamic systems is too basic and should be included in the problem setup at the start of the paper.
>
> Thanks for your comment. We have revised the paper based on your suggestions. The revised draft of Sec. 2 (Problem Definition) and Sec. 3.1 (Motivation and Framework Overview) is shown as below:
>
> # Sec. 2 Problem Setup
> Given a fluid dynamical system, we have $N$ sensors within the domain $\Omega$, with their locations denoted as $x\_1, \cdots, x\_N$, where $x_i \in \mathbb{R}^{d_l}$. The observations at time step $t$ are represented as $s_1^t, \cdots, s_N^t$, where $s_i^t \in \mathbb{R}^{d_o}$ and $d_o$ indicates the number of observation channels. Dynamical systems are governed by underlying system rules, such as PDEs with coefficient $\xi$. Variations in system parameters may lead to different environments, potentially resulting in distribution shifts. In our study, we are provided with historical observation sequences $\{s_i^{1: T_0}\}\_{i=1}^N$ and physical parameters $\xi$ (e.g., coefficients in the PDEs). Our goal is to predict the future observations of each sensor $s_i^{T_0+1: T_0+T}$. In dynamical systems, the out-of-distribution problem examines model performance when predicting under unseen parameter distributions or environments. Let $u^t=[s_1^t,\cdots, s_N^t]$, these systems evolve according to $\frac{d{u}}{dt} = F({u}, {\xi})$, where ${u}$ represents the observations and ${\xi}$ denotes the system parameters. When ${\xi} \sim P({\xi})$, the state trajectory ${u}^{1:T_0}$ follows the distribution $P({u}^{1:T_0} | {\xi})$. Assume we learn a learned mapping function $f$ from ${u}^{1:T_0}$ to ${u}^{T_0+1:T_0+T}$, i.e., ${u}^{T_0+1:T_0+T} = f({u}^{1:T_0})$ and there could be different distributions across training and test datasets, i.e., $P_{\text{train}}({\xi})\neq P_{\text{test}}({\xi})$, which results in $P\_{{train }}\left({u}^{1: T_0}\right) \neq {P}\_{{test }}\left({u}^{1: T_0}\right)$. Moreover, when conducting rollout prediction, we need to feed the output back to the model, i.e., ${u}^{T_{start}:T_{start}+T-1} = f({u}^{T_{start}-T_0:T_{start}-1})$, with $P({u}^{1:T_0} | {\xi})\neq P({u}^{T_{start}-T_0:T_{start}-1}|{\xi}, T_{start})$, which demonstrates temporal distribution shift.
>
> # Sec. 3 The Proposed PURE
> ## Sec. 3.1 Motivation and Framework Overview
>
> This paper addresses the challenge of out-of-distribution fluid system modeling, which is complicated by parameter-based and temporal distribution shifts. Specifically, our function $f(\cdot)$ can suffer from a serious distribution shift result from different ${\xi}$ and $T\_{start}$, i.e., $P({u}\_{input}|{\xi},T\_{start})$. To reduce the impact of distribution shift, we aim to learn invariant observation embeddings ${\mu}^t$ to different environments, i.e., ${\xi}$ and $T_{start}$ for better generalization and utilize prompt embeddings ${z}^t$ to indicate the current environment for final prediction. In formulation, we have:
>
> $$
> {z}^t \perp {\mu}^t, u_{\text {output }}=\phi\left(\left[{\mu}^t,  {z}^t\right]\right) . \quad (1)
> $$
>
> The first term ensures the invariance of observation embeddings by decoupling observation embeddings and prompt embeddings. The second term aims to combine both two embeddings to generate the future predictions. Therefore, we propose a novel approach named PURE as:
> $$
> {\mu}^t=\operatorname{BasicModel}\left({u}_{\text {input }}\right),  \quad (2)
> $$
>
> $$
> z^0=\operatorname{ContextMining}\left({u}_{\text {input }}\right),\quad (3)
> $$
>
> $$
> {z}^t=\operatorname{GraphODE}\left(z^0, t\right),\quad (4)
> $$
>
> where a basic model is adopted to generate observation, and we adopt context mining and graph ODE to learn time-varying prompt embeddings. Given a basic forecasting model (Eqn. 2), our PURE contains three key modules: (1) Multi-view Context Exploration, which explores spatio-temporal data using both the attention mechanism and the frequency domain to initialize prompt embeddings (Eqn. 3). (2) Time-evolving Prompt Learning, which incorporates the interpolation of observation sequences into a graph ODE to learn the evolution of prompt embeddings (Eqn.4). (3) Model Adaptation with Prompt Embeddings, which leverages the time-evolving prompts to mitigate the temporal distribution shifts in fluid dynamics models (Eqn.1). More details are in Figure 1.
>
> > Q2. It's essential to explain how the proposal's components address dynamic OOD, focusing on parameter invariance between training and testing.
>
> Thanks for your comment. We have included our motivation in Sec. 3.1.
>
> > Q3. Without revision, I can't connect the problem to the proposal from the rebuttal and submission exchanges.
>
> Thanks for your comment. We have revised the manuscript and included all the related content here for your convenience.
>
> Thank you again for your feedback! Please let us know if you have further questions.

---

> > ### Author Response · Authors · 2024-08-13
> > **Further clarification**
> >
> > Dear Reviewer,
> >
> > As the deadline for the author-reviewer discussion phase is approaching, we would like to check if you have any other remaining concerns about our paper. We greatly appreciate your feedback and have worked diligently to address your comments. Thanks to your suggestions, we have revised the draft. **For your convenience, we show all the revised part of our draft as follows** and we believe that it may not be necessary to review the rebuttal text and the submission back-and-forth at this time.
> >
> > > **Major concern about OOD challenge formulation and component necessity:**
> >
> >
> > We have revised the manuscript and included all the related content [Section 2 (Problem Setup) and Section 3.1 (Motivation and Framework Overview)] in our last response for your convenience.
> >
> > > **Clarification on the term "prompt":**
> >
> > We have revised the manuscript and included all the related content [Appendix C] below for your convenience.
> >
> > **Prompt Learning.** Prompt learning [46, 15, 23] has recently gained significant attention as a technique for adapting pre-trained models to various downstream tasks by leveraging the power of prompt-based fine-tuning [8, 27, 85, 69, 49]. In the domain of large language models, prompt learning aims to incorporate optimal tokens into the input sequence, which can effectively improve performance without extensive retraining [26, 83, 81]. In the context of fluid dynamics modeling, prompt learning refers to a supplementary hint to indicate the context, which is incorporated into the input (observation embedding) for better generalization. Although it shares a similar meaning as prompt tuning in language models, our prompt refers to the current environment, which determines the future evolution with better generalization.
> >
> > **[46]** Yajing Liu, Yuning Lu, Hao Liu, Yaozu An, Zhuoran Xu, Zhuokun Yao, Baofeng Zhang, Zhiwei Xiong, and Chenguang Gui. Hierarchical prompt learning for multi-task learning. CVPR2023
> >
> > **[15]** Yuxian Gu, Xu Han, Zhiyuan Liu, and Minlie Huang. PPT: Pre-trained prompt tuning for few-shot learning. arXiv2021
> >
> > **[23]** Tony Huang, Jack Chu, and Fangyun Wei. Unsupervised prompt learning for vision-language models. arXiv 2022
> >
> > **[8]** Ning Ding, Shengding Hu, Weilin Zhao, Yulin Chen, Zhiyuan Liu, Hai-Tao Zheng, and Maosong Sun. OpenPrompt: An open-source framework for prompt-learning. arXiv 2021.
> >
> > **[27]** Muhammad Uzair Khattak, Hanoona Rasheed, Muhammad Maaz, Salman Khan, and Fahad Shahbaz Khan. Maple: Multi-modal prompt learning. CVPR2023
> >
> > **[85]** Kaiyang Zhouet al. Conditional prompt learning for vision-language models. CVPR2022
> >
> > **[69]** Zifeng Wang et al.. Learning to prompt for continual learning.CVPR2022
> >
> > **[49]** Yuning Lu et al. Prompt distribution learning. CVPR2022
> >
> > **[26]** Woojeong Jin et al. A good prompt is worth millions of parameters: Low-resource prompt-based learning for vision-language models. arXiv 2021.
> >
> > **[83]** Zizhuo Zhang et al. Prompt learning for news recommendation. SIGIR 2023.
> >
> > **[81]** Yaohua Zha et al. Instance-aware dynamic prompt tuning for pre-trained point cloud models. CVPR2023
> >
> > > **Concern about Quantifying distribution shifts:**
> >
> > We have revised the manuscript and included all the related content [Appendix H] here for your convenience.
> >
> > **Performance with respect to Different Difficulty Levels.** Here, we demonstrate the performance of our PURE with varying difficulty levels. In particular, we measure the difficulty levels based on the distance between $P_{\text{train}}(\xi)$ and $P_{\text{test}}(\xi)$ and generate three levels on the Prometheus dataset. The compared results are shown in Table 7. From the results, we can observe that all the model performs worse in hard scenarios and our method has consistently outperformed these baselines. The potential reason is that (1) our model enhances model invariance across different distributions through decoupling prompt embeddings and observation embeddings via mutual information which results in high generalizationability to different environments; (2) our model utilizes multi-view context mining and graph ODE to extract prompt embeddings, which capture environment information accurately.
> >
> > Table 7. Performance comparison across varying levels of OOD generalization difficulty. The values represent the Mean Squared Error (MSE) for each method.
> > | Method      | U-NET  | ResNet | VIT    | Swin-T | FNO    | CGODE  | PURE(Ours) |
> > |-------------|--------|--------|--------|--------|--------|--------|------------|
> > | **Easy**    | 0.0945 | 0.0682 | 0.0654 | 0.0676 | 0.0452 | 0.0772 | *0.0325*    |
> > | **Mid**     | 0.1063 | 0.0922 | 0.0902 | 0.0912 | 0.0544 | 0.0863 |*0.0341*    |
> > | **Hard**    | 0.1432 | 0.1234 | 0.1076 | 0.1123 | 0.0623 | 0.0921 |  *0.0354*     |
> >
> > We hope that these revisions address your concerns. Please let us know if there are any further questions or concerns.
> >
> > Sincerely,
> > The Authors

---

### Official Review · Reviewer_MBSt · 2024-07-12

**Soundness:** 3
**Presentation:** 3
**Contribution:** 3
**Rating:** 5
**Confidence:** 5

**Summary:**

This paper pioneers the connection of prompt learning with dynamical system modeling to address the challenge of out-of-distribution shifts. The proposed PURE method initializes prompt embeddings by learning from historical observations and system parameters.

**Strengths:**

1.The paper is easy to follow.
2.The proposed method is sound and innovative.
3. The authors provide theoretical proof and show comprehensive experimental comparisons.

**Weaknesses:**

1. Some results may be incorrectly labeled as suboptimal in table, and there are errors in the use of some symbols.
2. The explanation of the experimental results is not detailed enough, making some experiments difficult to understand.
3. The proposed method is aimed at OOD (Out-Of-Distribution), but the experiments lack comparison and discussion with methods specifically targeting OOD, such as [1] and [2].
Reference:
[1] Kirchmeyer, Matthieu, et al. "Generalizing to new physical systems via context-informed dynamics model." International Conference on Machine Learning. PMLR, 2022.
[2] Yin, Yuan, et al. "LEADS: Learning dynamical systems that generalize across environments." Advances in Neural Information Processing Systems 34 (2021): 7561-7573.

**Questions:**

Q1. There might be misuses of symbols in the paper, such as, Change "xi" to "si" in line 65, Change "xq" to "xq" in line 96, "Zero-shot Experiments" and "Generalization Experiments" in line 215 should be treated equally and placed on separate lines.
Q2. Is there an issue with the second-best data in Table 2? For example, in the column w/OOD of SPHERICAL-SWE, the second-best should be DGPDE 0.0028. The corresponding improvement results also need to be modified.
Q3. What does the clustering in Figure 5 represent? Could you provide a detailed explanation?
Q4. The paper utilizes mutual information to decouple different prompt embeddings and observation embeddings, reducing the sensitivity of observation embeddings to different distributions. However, I don't quite understand the purpose of decoupling. Observational embeddings are related to the environment, and prompt embeddings are related to the environment as well; they are inherently correlated. Please provide further explanation.

**Limitations:**

This method does not apply to real-world scenarios, such as rigid dynamics modeling and traffic flow forecasting.

---

> ### Author Rebuttal · Authors · 2024-08-07
>
> We are truly grateful for the time you have taken to review our paper, your insightful comments and support. Your positive feedback is incredibly encouraging for us! In the following response, we would like to address your major concern and provide additional clarification.
>
>
>
> > Q1. Some results may be incorrectly labeled as suboptimal in table, and there are errors in the use of some symbols. There might be misuses of symbols in the paper, such as, Change "xi" to "si" in line 65, Change "xq" to "xq" in line 96, "Zero-shot Experiments" and "Generalization Experiments" in line 215 should be treated equally and placed on separate lines. Is there an issue with the second-best data in Table 2? For example, in the column w/OOD of SPHERICAL-SWE, the second-best should be DGPDE 0.0028. The corresponding improvement results also need to be modified.
>
>
> A1. Thank you for pointing this out. We will correct all the typos carefully in the revised version.
>
>
> > Q2. The explanation of the experimental results is not detailed enough, making some experiments difficult to understand. What does the clustering in Figure 5 represent? Could you provide a detailed explanation?
>
> A2. Thank you for your comment. We demonstrate the t-SNE visualization of the predicted trajectories for compared methods. From the results, we can observe that our method outperforms baselines, which validate the superiority of our method. Here, clustering refers to the technique of reducing the dimensions in t-SNE. We will include more details in our revised version.
>
>
>
> > Q3. The proposed method is aimed at OOD (Out-Of-Distribution), but the experiments lack comparison and discussion with methods specifically targeting OOD, such as [1] and [2]. Reference: [1] Kirchmeyer, Matthieu, et al. "Generalizing to new physical systems via context-informed dynamics model." International Conference on Machine Learning. PMLR, 2022. [2] Yin, Yuan, et al. "LEADS: Learning dynamical systems that generalize across environments." Advances in Neural Information Processing Systems 34 (2021): 7561-7573.
>
>
> A3. Thank you for your comment. We have added two baselines LEADS [1] and CODA [2] for performance comparison. The results below show that our method is superior to these baselines, which validates the effectiveness of our method when addressing OOD problems. We will include it in our revised version.
>
>
> | Dataset     | Prometheus ID | Prometheus OOD | ERA5 ID | ERA5 OOD | SSWE ID | SSWE OOD |
> |-------------|----------------|----------------|---------|----------|---------|----------|
> | LEADS       | 0.0374         | 0.0403         | 0.2367  | 0.4233   | 0.0038  | 0.0047   |
> | CODA        | 0.0353         | 0.0372         | 0.1233  | 0.2367   | 0.0034  | 0.0043   |
> | Ours | 0.0323         | 0.0328         | 0.0398  | 0.0401   | 0.0022  | 0.0024   |
>
>
>
>
> > Q4. The paper utilizes mutual information to decouple different prompt embeddings and observation embeddings, reducing the sensitivity of observation embeddings to different distributions. However, I don't quite understand the purpose of decoupling. Observational embeddings are related to the environment, and prompt embeddings are related to the environment as well; they are inherently correlated. Please provide further explanation.
>
> A4. Thank you for your comment. Our decoupling aims to build invariance of our observation embeddings to environments for better generalization. Here, both observation embeddings and prompt embeddings are combined for the final prediction, and prompt embeddings are utilized to provide the environment information. In other words, observational embeddings are not correlated with the environment for better generalization. We will include it in our revised version.
>
> > Q5. This method does not apply to real-world scenarios, such as rigid dynamics modeling and traffic flow forecasting.
>
> A5. Thank you for your comments.  We have added four baselines, i.e., EGNN [1], SGNN [2], SimVP [3], PastNet [4] for performance comparison on the RigidBall and TaxiBJ datasets. The table shows our method outperforms the baseline models in rigid dynamics and traffic flow forecasting. Please see the PDF for visual results. We will include it in our revised version.、
>
>
> **Rigid dynamics (MSE)**
>
> | Method | PL 10 $\downarrow$ | PL 20 $\downarrow$ | PL 30 $\downarrow$ | PL 40 $\downarrow$ | PL 50 $\downarrow$ |
> |--------|-------|-------|-------|-------|-------|
> | EGNN   | 1.37  | 1.89  | 3.77  | 5.66  | 7.87  |
> | SGNN   | 0.64  | 0.72  | 1.23  | 2.44  | 4.96  |
> | PURE   | **0.59** | **0.68**  | **0.97**  | **1.45**  | **3.99**  |
>
> **Traffic flow**
>
> | Metric | MSE $\downarrow$   | MAE $\downarrow$   | SSIM $\uparrow$ | PSNR$\uparrow$  |
> |--------|--------|--------|-------|-------|
> | SimVP  | 0.4332 | 16.897 | 0.9822| 39.29 |
> | PastNet| 0.4293 | 16.405 | 0.9876| 39.42 |
> | PURE   | **0.3982** | **15.434** | **0.9971**| **41.23** |
>
>
> **Reference**
>
> [1] Satorras, Vıctor Garcia, Emiel Hoogeboom, and Max Welling. "E (n) equivariant graph neural networks." ICML2021
>
>
> [2] Han, Jiaqi, et al. "Learning physical dynamics with subequivariant graph neural networks." NeurIPS2022
>
>
> [3] Gao, Zhangyang, et al. "Simvp: Simpler yet better video prediction." CVPR. 2022.
>
> [4] Wu, Hao, et al. "Pastnet: Introducing physical inductive biases for spatio-temporal video prediction." MM2024
>
> We will also add your suggestion about future works to our revised version. Thanks again for appreciating our work and for your constructive suggestions. Please let us know if you have further questions.

---

> ### Author Response · Authors · 2024-08-14
> **Summary of our rebuttal**
>
> Dear Reviewer,
>
> We summarized our rebuttal content at your convenience as follows:
> - We have included more competing baselines to demonstrate the superiority of our approach.
> - We have included more real-world settings such as rigid dynamics modeling and traffic flow forecasting for performance comparison.
> - We have explained the purpose of decoupling and visualization.
> - We will proofread our paper to clear every typo in our final version.
>
> We will also add your suggestion about future works to our revised version. Thanks again for appreciating our work and for your constructive suggestions. Please let us know if you have further questions.
>
> Best,
>
> the Authors

---

### Official Review · Reviewer_Sh2w · 2024-07-13

**Soundness:** 3
**Presentation:** 3
**Contribution:** 3
**Rating:** 7
**Confidence:** 2

**Summary:**

The paper proposes a graph ODE-based approach for OOD fluid dynamics modeling. PURE aims to learn time-evolving prompts via graph ODE for adaptation of spatio-temporal forecasting models on OOD scenarios. To address temporal distribution shifts, the interpolation of obersvation sequences are combined into graph ODE framework to learn evolution of prompt embeddings.

**Strengths:**

- The paper proposes a new approach that connects prompt learning and dynamical system modeling which addresses OOD shifts.
- By learning time-evolving prompts that adapt to changes in system parameters and temporal evolution, the approach can enhance model robustness.
- The paper provides theoretical analysis on incorporating observations during evolution.
- Experiments on diverse benchmarks show generalization ability to OOD and different prediction length.

**Weaknesses:**

As I am not an expert in this field, I am unable to find major concerns or weakness of the approach.
- As the method is based on attention, the proposed approach may have limited scalability and take long computation time. Is there a comparison on these with the previous works?

**Questions:**

Please address the questions in the Weaknesses.

**Limitations:**

The limitations are explained in Appendix I.

---

> ### Author Rebuttal · Authors · 2024-08-07
>
> We are truly grateful for the time you have taken to review our paper, your insightful comments and support. Your positive feedback is incredibly encouraging for us! In the following response, we would like to address your major concern and provide additional clarification.
>
> > Q1. As the method is based on attention, the proposed approach may have limited scalability and take long computation time. Is there a comparison on these with the previous works?
>
> A1. Thank you for your comment. We have added a comparison of computational costs below. From the results, we can observe that our method has a competitive computation cost with huge performance increasement. Without OOD, it improves performance by an average of 4.44%, and with OOD, it improves performance by an average of 25.90%. We will include it in our revised version.
>
> | Method       | UNet | ResNet | VIT   | SwinT | FNO  | UNO  | CNO  | NMO  | DGODE | PURE (Ours) |
> |--------------|-------|--------|-------|--------|------|------|------|------|-------|------------|
> | Training time (h) | 11.2  | 9.76   | 14.5  | 12.3   | 6.9  | 7.8  | 13.4 | 6.3  | 13.2  | 7.2        |
> | Inference time (s) | 1.34  | 0.93   | 1.32  | 1.13   | 0.54 | 0.67 | 0.12 | 0.52 | 1.23  | 0.69       |
>
> We will also add your suggestion about future works into our revised version. Thanks again for appreciating our work and for your constructive suggestions. Please let us know if you have further questions.

---

> > ### Comment · Reviewer_Sh2w · 2024-08-11
> >
> > Thank you for the additional experiments on computational costs.

---

> ### Author Response · Authors · 2024-08-11
> **Thank you for your feedback and support!**
>
> Thank you for your feedback and support! We will add the rebuttal contents to the main paper in the final version following your valuable suggestions.

---

### Author Rebuttal · Authors · 2024-08-07

Dear Reviewers,

Thanks for your time and valuable feedbacks. We acknowledge **three reviewers** (Reviewer Sh2w, Reviewer MBSt, and Reviewer MFD5) comments that **our work is novel or new**. We acknowledge the positive comments such as "a new approach" (Reviewer Sh2w), "enhance model robustness" (Reviewer Sh2w), "provides theoretical analysis " (Reviewer Sh2w), "show generalization ability" (Reviewer Sh2w), "easy to follow" (Reviewer MBSt), "sound and innovative" (Reviewer MBSt), "theoretical proof" (Reviewer MBSt), "comprehensive experimental comparisons" (Reviewer MBSt), "significant topic" (Reviewer vRkX), "well-motivated" (Reviewer vRkX), "the effective presentation" (Reviewer vRkX), "extensive experiments" (Reviewer vRkX), "novel idea" (Reviewer MFD5), "technically sound" (Reviewer MFD5), and "effectiveness" (Reviewer MFD5). We have also responded to your concerns in the following. The figures are included in the pdf file for your reference.

Please let us know if you have any additional questions or concerns. We will try our best to address them.

Best regards,

the Authors

---

### Author Response · Authors · 2024-08-14
**Summary of Rebuttal**

Dear Reviewers and Area Chairs,

We would like to express our sincere gratitude for your great efforts, insightful comments, support, and the constructive suggestions you have provided once again! Through our discussions and the reviewers' responses, it appears that we have effectively addressed the major concerns raised by reviewers. This outcome has greatly benefited us, and we would like to thank all of you for your valuable support!

The reviewers held many positive comments on our paper. For example, **three reviewers** (Reviewer Sh2w, Reviewer MBSt, and Reviewer MFD5) commented that our work is **novel or new**. Reviewer MFD5 and Reviewer MBSt commented that the paper is **technically sound**. Reviewer MBSt acknowledged that our paper is **easy to follow** and shows **comprehensive experimental** comparisons. Reviewer vRkX believes that our work is **well-motivated**.

The reviewers also raised insightful and constructive concerns. We made every effort to address all the concerns by providing detailed clarification and requested results. Here is the summary of the major revisions:

- We have **clarified our problem definition and motivation**, and provided the corresponding **revised draft** for reviewers' convenience.
- We have added **efficiency analysis** to make the paper more complete.
- We have included **more competing baselines** including LEADS, CODA and NUWA to demonstrate the superiority of our approach.
- We have demonstrated **more performance comparisons** in different real-world scenarios to validate the effectiveness of the proposed method in the real world.

We firmly believe that our framework for out-of-distribution fluid dynamics modeling plays a significant role in advancing the community. And we are committed to making our complete code and training details publicly available. All the rebuttal contents will be properly included in the final version, following your valuable suggestions. Moreover, we are eager to engage in further discussions with you to enhance our understanding of the domain and further improve the quality of the paper.

Once again, thank you for your time and effort in reviewing our work. We greatly appreciate your assistance in improving our manuscript!

Best regards,

the Authors

---

### Decision · Program_Chairs · 2024-09-25

**Decision:**

Accept (poster)

**Comment:**

This paper proposes a new approach for out-of-distribution fluid dynamics modeling, PURE. It can learn time-evolving prompt embedding, using a graph ODE, for model adaptation, to overcome temporal distribution. Experiments show its effectiveness.

Firstly, the paper studies an important topic in fluid dynamics modeling. Secondly, the idea is new, and the techniques are sound. Finally, the paper provides extensive empirical results to support its claims. However, the presentation is not clear enough. For example, the studied problem is not well-formulated. And there are quite a few typos.

The authors should further improve the paper carefully!